



# Source apportionment of carbonaceous chemical species to fossil fuel combustion, biomass burning and biogenic emissions by a coupled radiocarbon-levoglucosan marker method

Imre Salma[1,2], Zoltán Németh[1,2], Tamás Weidinger[3], Willy Maenhaut[4], Magda Claeys[5],
Mihály Molnár[6], István Major[6], Tibor Ajtai[7], Noémi Utry[7], Zoltán Bozóki[7]

[1]Institute of Chemistry, Eötvös University, H-1518 Budapest, P.O. Box 32, Hungary
[2]Excellence Center, Faculty of Science, Eötvös University, H-2462 Martonvásár, Brunszvik u. 2, Hungary
[3]Department of Meteorology, Eötvös University, H-1518 Budapest, P.O. Box 32, Hungary
[4]Department of Analytical Chemistry, Ghent University, Krijgslaan 281, B-9000 Gent, Belgium
[5]Department of Pharmaceutical Sciences, University of Antwerp, Universiteitsplein 1, B-2610 Antwerp, Belgium
[6]Hertelendi Laboratory of Environmental Studies, Isotope Climatology and Environmental Research Centre,
 Institute of Nuclear Research, Bem tér 18/c, H-4026 Debrecen, Hungary
[7]MTA-SZTE Research Group on Photoacoustic Spectroscopy, University of Szeged, Dóm tér 9, H-6720 Szeged, Hungary

*Correspondence to*: Imre Salma (salma@chem.elte.hu)

**Abstract.** An intensive aerosol measurement and sample collection campaign was conducted in central Budapest in a mild winter for two weeks. The on-line instruments included a FDMS-TEOM, Aethalometer, photoacoustic spectrometer, RT-OC/EC analyser, DMPS, gas pollutant analysers and meteorological sensors. The aerosol samples were collected on quartz fibre filters by a low-volume sampler using the tandem filter method. Elemental carbon (EC), organic carbon (OC), levoglucosan, mannosan, galactosan, arabitol and mannitol were determined, and radiocarbon analysis was performed on the aerosol samples. Median atmospheric concentrations of EC, OC and $PM_{2.5}$ mass were 0.97, 4.9 and 25 μg m$^{-3}$, respectively. The EC and organic matter (1.6×OC) accounted for 4.8% and 37%, respectively of the $PM_{2.5}$ mass. Fossil fuel (FF) combustion represented 36% of the total carbon (TC=EC+OC) in the $PM_{2.5}$ size fraction. Biomass burning (BB) was a major source (40%) for the OC in the $PM_{2.5}$ size fraction, and a substantial source (11%) for the $PM_{10}$ mass. We proposed and applied here a novel, straightforward, coupled radiocarbon-levoglucosan marker method for source apportionment of the major carbonaceous chemical species. The contributions of EC and OC from FF combustion ($EC_{FF}$ and $OC_{FF}$) to the TC were 11.0% and 25%, respectively, EC and OC from BB ($EC_{BB}$ and $OC_{BB}$) were responsible for 5.8% and 34%, respectively of the TC, while the OC from biogenic sources ($OC_{BIO}$) made up 24% of the TC. Evaluation of the apportioned atmospheric concentrations revealed some of their important properties and relationships among them. $EC_{FF}$ and $OC_{FF}$ were associated with different FF combustion sources. Most $EC_{FF}$ was emitted by vehicular road traffic, while the contribution of non-vehicular sources such as domestic and industrial heating or cooking using gas, oil or coal to $OC_{FF}$ was substantial. The mean contribution of BB to soot particles was smaller by a factor of approximately 2 than of road traffic. The main formation processes of $OC_{FF}$, $OC_{BB}$ and $OC_{BIO}$ from volatile organic compounds were jointly influenced by a common factor, which is most likely the atmospheric photochemistry, while primary organic emissions can be also important. Technological improvements and control measures for various BB appliances, together with efficient enlightenment of their users, in particular on the admissible fuel types offer important potentials for improving the air quality in Budapest, and likely in other cities in general.

## 1 Introduction and objectives

Carbonaceous chemical species are abundant and important components of atmospheric aerosol particles in urban, rural and continental background environments from many aspects (e.g. Fuzzi et al., 2015 and references therein). They can affect the climate, air quality, visibility, human health, ecosystems and built environment on local, regional and sometimes even on larger (global) spatial scales. Their major source types include both primary emissions and secondary particle formation processes, which are related mainly to fossil fuel (FF) combustion, biomass burning (BB) and biogenic emissions. Characterisation of





particles from various sources and quantification of the contributions of the sources are necessary to understand the role and impacts of atmospheric aerosol in general and specifically in cities as well. Moreover, various forms of BB, in particular wood burning in household appliances for heating, cooking or pleasure, in boilers and industrial power plants are expected to rise in the coming years, which can lead to enlarged concentrations of some groups of organic molecules and particulate matter (PM)

mass on shorter (seasonal) time scales or as a tendency (Saarikoski et al., 2007; Gilardoni et al., 2011; Saarnio et al., 2012; Bernardoni et al., 2013; and references therein). The large number, complex character, spatial and temporal variability of the emission and formation sources of carbonaceous chemical species together with their dynamically changing transformation processes and atmospheric conditions make the quantification of the source types or their inventory-based source assessment challenging. There are, however, some receptor models, which facilitate the source apportionment on the basis of atmospheric

concentrations and some model-derived properties. These include the source-specific marker methods, the so-called Aethalometer model (based on the wavelength dependence of the optical absorption coefficient, and thus, it is not confined only to the Aethalometer data) and various multi-statistical models. The marker methods - such as the radiocarbon or levoglucosan methods - are advantageous from the point of view that they do not require many samples or extensive data sets, are quite straightforward, and, therefore, they are often the choice for source apportionment studies.

Biomass burning produces a big variety of organic molecules. Of them, three monosaccharide anhydrides, namely levoglucosan (LVG, 1,6-anhydro-β-D-glucopyranose, $C_6H_{10}O_5$; Simoneit et al., 1999) and its stereoisomers mannosan (MAN, 1,6-anhydro-β-D-mannopyranose) and galactosan (GAN, 1,6-anhydro-β-D-galactopyranose; Nolte et al., 2001) are abundant organic species in the aerosol phase. They are formed during pyrolysis of the bulk materials of wood such as cellulose and

hemicellulose at temperatures larger than 300 °C (Caseiro et al., 2009), and are not present in the vapour phase. Levoglucosan, which is the most abundant of them (Simoneit et al., 1999), is considered reasonably stable in the atmosphere due to photolysis and acid-catalysed hydrolysis for at least 10 d (Locker, 1988; Fraser and Lakshmanan, 2000; Simoneit et al., 2004). Its lifetime can be, however, decreased by chemical reactions with OH radical in the aqueous phase under high relative humidities (Hennigan et al., 2010; Hoffmann et al., 2010). Such conditions can be important in tropical areas or for long-range transported,

aged smoke plumes in summer. Nevertheless, LVG was regarded to be a conservative molecular marker in most studies on BB (Simoneit et al., 1999; Zdráhal et al., 2002; Puxbaum et al., 2007; Saarikoski et al., 2008; Szidat et al., 2009; Maenhaut et al, 2012a). Moreover, the ratios of the stereoisomers can indicate the relative proportion of hardwood and softwood burning. Levoglucosan can also be produced in the pyrolysis of lignite (Fabbri et al., 2009) and peat (Iinuma et al., 2007; Kourtchev et al., 2011), which can complicate the apportionment procedure.


There are additional organic markers that can be determined jointly with the monosaccharide anhydrides, and which can supply useful information on some bioaerosols. These are two sugar alcohols, namely arabitol (ARL, (2R,4R)-pentane-1,2,3,4,5-pentol, $C_5H_{12}O_5$) and mannitol (MAL, (2R,3R,4R,5R)-hexane-1,2,3,4,5,6-hexol, $C_6H_{14}O_6$), which ordinary originate from metabolic activity of fungi and bacteria (Bauer et al., 2008; Burshtein et al., 2011; Gosselin et al., 2016). Fungi are important

microorganisms because they contribute substantially to the decomposition of organic material. Most fungi emit spores into the air. The fungal spores can irritate the respiratory system, cause allergies or infectious acute diseases, and chronic sicknesses, in particular in indoor environments. Their presence in the air is commonly quantified by the spore counts method, which provides their contribution in terms of particle number. The spores contain ARL and MAL as storage substances. Correlation between the fungal spore count and their concentration was verified in the $PM_{10}$ size fraction (Bauer et al., 2008; Zhang et al.,

2010). Mannitol is one of the common energy and carbon storage molecules produced by various organisms, including bacteria, yeasts, fungi, algae, lichens, and many plants but its major suspension into the air is often expected to be linked primarily to fungal spores under ordinary atmospheric conditions.





The radiocarbon method makes it feasible to distinguish between the carbon originating from fossil and non-fossil (contemporary) sources by determining the isotopic ratios of C (Szidat et al., 2006 and references therein). Secondary neutrons generated by cosmic radiation in the upper atmosphere interact with atmospheric N and produce radioactive $^{14}C$ with a half-life and standard deviation (SD) of 5730±40 y. This radionuclide is taken up by the living biosphere mainly via photosynthesis

and the food chain, which results in contemporary isotopic abundance of $^{14}C$ in the biomass. The FF formation in buried dead organisms detached from the atmospheric interactions takes $>10^{7-8}$ y, during which the $^{14}C$ content decays and becomes negligible in FFs. The radiocarbon measurement results are usually expressed as the $^{14}C/^{12}C$ isotope ratios in the samples relative to that for the unperturbed atmosphere in the reference year of 1950 (Burr and Jull, 2009). Source apportionments based solely on this method usually take advantage of the fact that elemental carbon (EC) is introduced into the atmosphere

either from FF combustion or BB exclusively as primary particles. The apportionment is achieved by determining the C isotopic ratio specifically for EC (and organic carbon, OC) separated by different thermal treatment (and later also for other carbonaceous fractions such as water soluble or water insoluble OC) and combining this information with atmospheric concentrations and emission factors (e.g. Szidat et al., 2006, 2009; Minguillón et al., 2011; Bernardoni et al., 2013).

In earlier studies in Budapest, the mean contribution of organic matter (OM, OM=1.4×OC, where 1.4 is the OM/OC mass conversion factor, see Sect. 2.2) to the $PM_{2.5}$ mass was 43% for both a street canyon (kerbside) in the city centre and near-city background (Salma et al., 2004; Maenhaut et al., 2005). Elemental carbon made up on average 14% of the $PM_{2.5}$ mass in the street canyon, while its contribution in the near-city background was much smaller, 2.1%. Source apportionment of C has not been assessed so far. There has been an increasing and definite need to determine the tendencies in the concentration levels,

abundances of the major carbonaceous species, and, in particular, to quantify the contribution and relevance of FF combustion, biogenic emissions and BB in cities all over Europe. These goals can be achieved by combining several on-line and off-line experimental methods. For the source apportionment, we propose here a coupled straightforward approach based on both radiocarbon and LVG marker methods. The main objectives of the present paper are to demonstrate the potentials of the analytical data set derived by several methods which are based on different principles and which yield data with various time

resolutions, to quantify the contributions of FF combustion, BB and biogenic sources by the coupled radiocarbon-LVG marker method for a winter season, to study the properties of and relationship among the apportioned carbonaceous species, to interpret their consequences on the air quality in Budapest as a Central European city, and to discuss the details and potentials of wood burning in the area.

## 2 Methods

### 2.1 Measurement and sample collection campaign

On-line aerosol measurements and collection of aerosol samples were performed at the Budapest platform for Aerosol Research and Training (BpART) facility in Budapest (Salma et al., 2016). The site represents a well-mixed, average atmospheric environment for the city centre. The sampling inlets and sensors were set up at heights between 12 and 13 m above the street level of the closest road. The aerosol campaign took place continuously for two weeks from Tuesday, 25–02–

2014 to Tuesday, 10–03–2014. Calm weather situations were present all over the campaign; milder than ordinary winter air temperatures occurred, and there was no snow cover in the region at all.

The on-line aerosol instruments included 1) a tapered element oscillating microbalance with a filter dynamics measurement system (FDMS-TEOM 1400a, Rupprecht and Patashnick, USA) for the PM mass; 2) a semi-continuous OC and EC analyser

(RT-OC/EC analyser, Sunset Laboratory, USA); 3) an Aethalometer (AE33, Magee Scientific, USA) for optical attenuation of aerosol samples collected on Teflon-coated glass fibre filter tape spots at seven wavelengths of 370, 470, 520, 590, 660,



880 and 950 nm; 4) a photoacoustic spectrometer (PAS) for the optical absorption coefficient (OAC) of particles in their dispersed state at four wavelengths of 266, 355, 532 and 1064 nm (Ajtai et al., 2010, 2015); 5) a differential mobility particle sizer (DMPS, Salma et al., 2011) for measuring particle number size distribution in a diameter range of 6–1000 nm; and 6) a LI-840 $CO_2$ analyser with a single path, dual wavelength non-dispersive infrared detection system (LI-COR, USA). The

aerosol sampling inlets contained upper-size sharp-cut cyclones (URG, USA) with a 50%-efficiency at an aerodynamic diameter of 2.5 µm. Pallflex Tissuquartz filters (Pall, USA) were used in the RT-OC/EC analyser, and the EUSAAR2 thermal protocol (He gas: 200 °C for 120 s, 300 °C for 150 s, 450 °C for 180 s, and 650 °C for 180 s; mixture of 2% $O_2$ in He: 500 °C for 120 s, 550 °C for 120 s, 700 °C for 70 s, and 850 °C for 80 s; Cavalli et al., 2010) was selected for the measurements. This protocol is recommended for urban samples. The time resolution ($\tau$) of the AE33 and PAS instruments were 1 min and 18 s,

respectively, and it was ca. 10 min for the FDMS-TEOM and DMPS systems. The RT-OC/EC analyser typically collected samples for 2 h 45 min, while the analysis took place for 15 min, which yielded measured data every 3 h. $CO_2$ was measured with $\tau$=1 min. The concentration of some criteria pollutant gases was obtained from the closest measurement station of the National Air Quality Network in Budapest at a distance of 1.6 km from the BpART facility in the upwind prevailing wind direction. $SO_2$, $O_3$ and $NO_x$ were determined by UV fluorescence (Ysselbach 43C), UV absorption (Ysselbach 49C) and

chemiluminescence (Ysselbach 42C) methods, respectively with $\tau$=1 h. Basic meteorological data including air temperature outside and inside the BpART ($T_{out}$ and $T_{in}$, respectively), relative humidity outside and inside the platform ($RH_{out}$ and $RH_{in}$, respectively), global solar radiation (GRad), wind speed (WS) and wind direction (WD) were measured by an on-site meteorological station with $\tau$=10 min.

The aerosol samples were collected by using a low-volume (1 $m^3$ $h^{-1}$), Gent-type stacked filter unit (SFU) sampler (Maenhaut et al., 1994). The collection device was loaded with a coarse Nuclepore filter in the first stage, and two, front and back Pallflex Tissuquartz quartz fibre filters directly on top of each other in the second stage. All three filters had a diameter of 47 mm. The quartz filters had the same manufacturer lot (batch) number to ensure their identical adsorption properties, and had been pre-baked at a temperature of 550 °C for 12 h prior to sampling to remove possible organic contaminants. The Nuclepore filters

and front quartz filters collect $PM_{10-2.5}$ and $PM_{2.5}$ particles, respectively. A total of 14 exposed filter sets for the daylight periods (from about 6:30 to 18:20 local time, UTC+1), and 14 exposed filter sets for the nights (from about 18:30 to 6:20) together with two field blank sets were obtained. The filters were placed into polycarbonate Petri slide dishes, and were stored frozen until analyses.

## 2.2 Analyses of aerosol samples and data treatment

The PM mass concentrations were obtained by weighing each Nuclepore and front quartz filter twice before and twice after sampling on a microbalance with a sensitivity of 1 µg. The filters were pre-equilibrated before weighing at a temperature of ca. 20°C and RH of 50% for at least 24 hours. The gravimetric data for the real exposed filters were corrected for the net PM mass using the field blank filters. The mean blank masses for the Nuclepore and front quartz filters corresponded to 4.0 and 8.0 µg $m^{-3}$, respectively. One or two punches with an area of 1.5 $cm^2$ of the quartz filters were analysed by the thermal-optical

transmission (TOT) method (Birch and Cary, 1996) by a laboratory OC/EC analyser (Sunset Laboratory, USA) using the NIOSH2 thermal protocol (He gas: 310 °C for 60 s, 480 °C for 60 s, 615 °C for 60 s, and 870 °C for 90 s; mixture of 2% $O_2$ in He: 550 °C for 45 s, 625 °C for 45 s, 700 °C for 45 s, 775 °C for 45 s, 850 °C for 45 s, and 870 °C for 120 s). This protocol was selected for comparative reasons since it had been also employed for our earlier studies in Budapest (Salma et al., 2004; Maenhaut et al., 2005). All measured OC and EC data for the front filters were above the determination limits. The overall

relative uncertainty of the TOT analysis was estimated to be 5%+0.2 µg C $cm^{-2}$ for both OC and EC (Viana et al., 2006). The adsorptive sampling artefacts of the organic constituents were corrected by subtracting the concentration of OC for the back





quartz filters from the corresponding OC data for the front quartz filters according to the tandem filter method (Kirchstetter et al., 2001 and references therein). The back/front concentration ratios for the blank-corrected data ranged from 1.7% to 48% with a mean and SD of 22±13%. Elemental carbon was near or below the determination limit on the back quartz filters, with a mean back/front ratio and SD of 5.5±5.6%. For this reason, no correction for sampling artefact was adopted for EC. In order

to convert the concentrations of OC into OM, the OC data were multiplied by an OM/OC conversion factor of 1.6, which was suggested for oxidizing urban environments (Turpin and Lim, 2001; Russell, 2003). It was estimated that the relative uncertainty associated with the conversion is approximately 30% (Maenhaut et al., 2012a). A filter section with an area of 1.7 cm$^2$ of each front quartz filter was also analysed for LVG, MAN, GAN, ARL and MAL by gas chromatography/mass spectrometry (GC/MS) after extraction and trimethylsilylation using a modified method of Pashynska et al. (2002). The

extraction was now done with methanol. The recovery standard in the present work was methyl O-L-xylanopyranoside. The GC temperature program was also slightly modified to the following: an initial temperature of 100 °C was maintained for 2 min, it was followed by a gradient of 3 °C min$^{-1}$ to 200 °C, with the latter kept constant for 2 min, then followed by a gradient of 30 °C min$^{-1}$ to 310 °C, after which this temperature was preserved for 2 min. The monosaccharide anhydrides and sugar alcohols were obtained above the determination limit in all samples, while they were not measured on the back quartz filters.

Three-quarter sections of the front and back quartz filters were subjected to well-maintained C isotope analysis of the total carbon (TC=EC+OC) content by using accelerator mass spectrometry (AMS). The filter sections were treated in an off-line combustion system, which was designed specifically for this purpose (Molnár et al., 2013). The samples were placed in test tubes together with 15 mg $MnO_2$ and 5 mg Ag wool reagents, and the tubes were evacuated to vacuum (<5×10$^{-8}$ bar). The

carbonaceous compounds were oxidised quantitatively to $CO_2$ gas by the $MnO_2$ at a temperature of 550 °C for 3 days. The $CO_2$ gas was cryogenically separated from the other combustion gases and water vapour, and it was purified in a dedicated vacuum line. The amount of $CO_2$ was determined by using a high-precision pressure measurement. The sample preparation yield was calculated from the C mass derived by the pressure measurement and the uncorrected TC obtained from the laboratory OC/EC analyser. The $CO_2$ gas samples containing 20–150 µg C were introduced into a Mini Carbon Dating System

spectrometer (Enviro-MICADAS, IonPlus, Switzerland) via its dedicated gas ion source interface with He carrier gas at a constant flow rate. The field blank filters were prepared identically to the front filters. In addition to the aerosol and field blank filters, several procedure blank samples were also prepared by filling the test tube with fossil $CO_2$ gas and by following an identical sample preparation treatment as for the filters in order to determine the analytical procedure blank value for the AMS data. Based on these experiments, a mean analytical procedure blank correction factor and SD of 1.0±0.1 µg modern C (see

below) per sample was obtained, and it was adopted for all aerosol filters. The $^{14}C/^{12}C$ ratios were also corrected for isotopic fractionation by using the $^{13}C/^{12}C$ ratios (Wacker et al., 2010) that were obtained simultaneously in the actual AMS measurements. The $^{14}C/^{12}C$ isotopic ratios derived were also normalised to that of the oxalic acid II standard reference material (NIST 4990C, USA), and the measurement results were expressed as fraction of modern carbon ($f_M$), which denotes the $^{14}C/^{12}C$ ratios of the samples relative to that of unperturbed atmosphere in the reference year of 1950 (Burr and Jull, 2009). As the

majority of the currently combusted firewood was growing during the interval of atmospheric nuclear fusion bomb tests in the late 1950s and early 1960s, the aerosol particles originating from recent wood contain higher radiocarbon than corresponds to the present atmosphere by a mean factor of 1.08 for the Northern Hemisphere (Szidat et al., 2009; Heal et al., 2011). Thus, the fraction of contemporary carbon ($f_C$) was calculated as $f_C=f_M/1.08$, while the remaining fraction of the TC was regarded to be the fraction of fossil carbon ($f_{FF}=1-f_C$).


The FDMS-TEOM resulted in ca. 2 thousand data lines during the aerosol campaign. 99% of the base mass concentrations were >5 µg m$^{-3}$, which is the determination limit of the method. The reference mass concentration, which represents the correction for semi-volatile chemical species and water vapour, varied from –10.3 to 2.0 µg m$^{-3}$ with a median of –3.3 µg m$^-$





[3]. It corresponds to a median correction factor of 15% in absolute value, which is in line with previous data. The sum of the $PM_{2.5}$ mass derived by the FDMS-TEOM and the $PM_{10-2.5}$ mass obtained from the SFU sampler was considered as the $PM_{10}$ mass. The AE33 measurements yielded ca. 20 thousand data lines. The instrument automatically adopts the dual spot method (Weingartner et al., 2003; Virkkula et al., 2007) for correcting the filter loading effect. The black carbon (BC_AE)

concentrations were derived from the measurements at a wavelength of 880 nm by using a mass absorption coefficient MAC=16.6 $m^2$ $g^{-1}$. The PAS instrument measured the background for 6 min in each hour and the ambient aerosol in the rest of the time, which resulted in ca. 53 thousand data lines. The BC_PAS concentrations were derived from the measured OACs at a wavelength of 1064 nm by assuming an Ångström exponent of 1.00 in the dependency of the OAC on wavelength. A MAC value of 16.6 $m^2$ $g^{-1}$ used in the AE33 for 880 nm was extrapolated to the wavelength of 1064 nm at which value the

PAS measures the BC. This yielded a MAC=13.7 $m^2$ $g^{-1}$, which was utilised in the concentration calculations.

Movement of the air masses was assessed by backward trajectories, which were generated by using the air parcel trajectory model HYSPLIT v4.9 with an option of vertical velocity mode (Draxler and Rolph, 2013). The Embedded Global Data Assimilation System meteorological database was utilised for the modelling. Trajectories arriving at the receptor sites at a

height of 200, 500, 2300 m above the ground level at 6:00 and 18:00 local time were calculated.

### 3 Results and discussion

Data validation and its conclusions for the experimental methods are to be discussed in a separate article. It is just noted here that the concentration of soot particles obtained by the different measuring methods were evaluated briefly by their correlation coefficients, mean concentration ratios and SDs for the 2×14 collection time periods (Table 1). The mean concentration ratio

between the EC_TOT and BC_PAS data sets was the closest value to 1.00 (0.87), while the correlation coefficient between the EC_TOT and BC_AE data sets was the largest ($r$=0.923). The BC data derived by the AE were larger than both the EC_TOT and BC_PAS data sets by similar factors of 2.9 and 3.3, respectively. These relatively large differences can be explained by the basic limitation of the Aethalometer, namely that unknown multiple scattering of light in the filter causes a longer optical path, which results in overestimating the concentrations of soot particles (Drinovec et al., 2015). The

enhancement factor is often between 2 and 6, and depends in a complex way on various parameters such as the filter material, amount of scattering particles embedded in the filter, the mixing state of absorbing and scattering particles, the face velocity of the sample air flow and RH. The EC concentration data obtained by the RT-OC/EC analyser ($\tau$=3 h) were larger than that determined by laboratory OC/EC analyser, while the corresponding OC data were smaller. This is in part explained by the difference in thermal protocol used in the two instruments (Maenhaut et al., 2012b; Panteliadis et al., 2015); the more

sophisticated instrumentation (e.g. detector) and larger amounts of TC in the laboratory OC/EC analyser may also have contributed. It has to be further noted that the time intervals for the RT-OC/EC data were not exactly the same as the sample collection periods. The EC_TOT data set ($\tau$=12 h) was accepted to characterise the absolute concentration level of soot particles, the BC_PAS ($\tau$=18 s) data were regarded to represent their temporal variability and partially, their absolute concentrations, while the BC_AE ($\tau$=1 min) data set represented their relative changes in time (time variability). It is also

mentioned that the MAC value used is not generally valid but can vary with aerosol composition, and morphology, origin and age of soot particles, which can further complicate the measurements.





**Table 1.** Correlation coefficients (upper diagonal triangle matrix) and mean concentration ratios with SD (lower diagonal triangle matrix) of EC measured by laboratory OC/EC analyser (EC_TOT), BC derived by AE33 (BC_AE) and by PAS (BC_PAS), and EC obtained by RT-OC/EC analyser (EC_RT-TOT). Averaging of the on-line data was performed for the sample collection time interval of the SFU sampler. The correlation coefficients and mean concentration ratios express the quantities in the line-to-column order.

| Variable | EC_TOT | BC_AE | BC_PAS | EC_RT-TOT |
|----------|--------|-------|--------|-----------|
| EC_TOT | 1.00 | 0.923 | 0.762 | 0.708 |
| BC_AE | 2.9±0.4 | 1.00 | 0.803 | 0.832 |
| BC_PAS | 0.87±0.11 | 0.30±0.05 | 1.00 | 0.677 |
| EC_RT-TOT | 2.1±0.6 | 0.75±0.15 | 2.5±0.7 | 1.00 |

### 3.1 Averages

The individual on-line concentrations and meteorological data were averaged for the 2×14 sampling time periods. Their ranges, overall medians, means with SDs are shown in Table 2 together with the atmospheric concentrations of chemical species obtained from the SFU filters. The concentrations observed are consistent with those previously reported for urban environments in Europe (Putaud et al., 2010). The aerosol data are also in line with the decreasing tendency in the PM mass, OC and EC identified for the city centre of Budapest for the beginning of 2000's years (Salma et al., 2004; Salma and Maenhaut, 2006). The EU 24-h health limit value for $PM_{10}$ mass of 50 μg m$^{-3}$ was exceeded three times, on 25, 26 and 27 February (on the first three days of the campaign). The aerosol particle number and pollutant gas concentrations correspond to ordinary levels in central Budapest (Salma et al., 2016), and the meteorological data indicated calm weather situations without extremes, but milder air temperatures than typically present in this time were observed. The monosaccharide anhydrides and sugar alcohols were now determined in Budapest for the first time. The median concentrations of LVG, MAN and GAN are comparable to other urban sites in Europe in winter (Szidat et al., 2009 and references herein; Maenhaut et al., 2012a). They exhibit pronounced seasonal variation with a maximum in winter followed by autumn, spring and summer (Caseiro et al., 2009; Kourtchev et al., 2011; Maenhaut et al., 2012a), which indicates that BB preferably happens in the coldest months. During winter in Europe, residential wood burning is the major source of LVG, and the observed concentrations are typically <1 μg m$^{-3}$ (Claeys et al., 2010; Caseiro and Oliveira, 2012; Herich et al., 2014; Yttri et al., 2015). Levoglucosan was the most abundant monosaccharide anhydride with a mean contribution and SD of 90±1%, followed by MAN and GAN with corresponding values of 6.3±1.0% and 3.7±0.4%, respectively. The average concentrations of ARL and MAL were somewhat smaller than those reported for other locations. Arabitol and MAL in Vienna, Austria, during the autumn varied between 7 and 63 ng m$^{-3}$, and between 8.9 and 83 ng m$^{-3}$, respectively (Bauer et al., 2008), and their mean concentrations in Rehovot, Israel, in winter were 8.4 and 22 ng m$^{-3}$, respectively (Burshtein et al., 2011). The differences can likely be explained by the variations in the types of fungus species, different climate and vegetation. Arabitol and MAL usually show considerable monthly variability with higher concentrations during autumn, and low levels during winter for ARL and summer for MAL. It was shown that an established biomarker for fungi (ergosterol) correlated with ARL and MAL only during the spring and autumn. This correlation might be related to high levels of vegetation during spring blossoms and autumn decomposition, and not necessarily have a direct relation with fungi levels (Burshtein et al., 2011).

There was an indication of larger $PM_{10–2.5}$ mass, total particle number concentration ($N$) and ultrafine (UF) concentration during daylight periods than for nights, while the $PM_{2.5}$ mass, $PM_{2.5}/PM_{10–2.5}$ mass ratio, OC and LVG exhibited larger values for nights than for daylight periods. The EC and BC data did not seem to show tendentious daytime variability. The mean contribution and SD of $PM_{2.5}$ mass to the $PM_{10}$ were 56±11% during the daylight times, while they were 63±11% during the nights. This is different from earlier results when the share of the $PM_{10–2.5}$ mass was larger than of $PM_{2.5}$ (Salma et al., 2001, 2004; Salma and Maenhaut, 2006). The night-to-daylight period ratio for LVG was 1.41 for the median concentrations. These





all can be associated with the diurnal pattern of major urban sources for these chemical species, by relatively short atmospheric residence time of both $PM_{10-2.5}$ particles and UF particles (the latter make up 75–90% of $N$ in Budapest; Salma et al., 2014), by relatively long atmospheric residence time for the $PM_{2.5}$ particles (which include soot particles, and likely contain large mass fractions of OC), and by diurnal cycling of some meteorological properties, in particular of GRad, planetary boundary

layer height and atmospheric mixing intensity.

**Table 2.** Range, median, mean with SD of atmospheric concentrations for $PM_{2.5}$ mass obtained by FDMS-TEOM, $PM_{10-2.5}$ mass obtained from the SFU sampler, $PM_{10}$ mass as the sum of the previous two on a sample-by-sample basis, BC derived by AE33 (BC_AE) and by PAS (BC_PAS), EC and OC measured by RT-OC/EC TOT analyser (EC_RT-TOT and OC_RT-TOT, respectively) and laboratory OC/EC TOT method (EC_TOT and OC_TOT, respectively), levoglucosan, mannosan, galactosan, sum of the three monosaccharide anhydrides ($\Sigma$MAs),
arabitol, mannitol, total aerosol particle number concentration ($N$), ultrafine particle number concentration (UF), $SO_2$, $O_3$, $NO_x$, NO and $CO_2$ concentrations, and air temperature and RH outside ($T_{out}$, $RH_{out}$, respectively) and inside ($T_{in}$, $RH_{in}$, respectively) the BpART research facility, and wind speed (WS). The averaging of the on-line data was performed for the sample collection time periods of the SFU sampler.

| Variable | Unit | Min | Median | Max | Mean | SD |
|---|---|---|---|---|---|---|
| $PM_{2.5}$ | µg m$^{-3}$ | 11 | 25 | 47 | 25 | 10 |
| $PM_{10-2.5}$ | µg m$^{-3}$ | 8.1 | 15.9 | 25 | 15.6 | 4.4 |
| $PM_{10}$ | µg m$^{-3}$ | 16 | 37 | 68 | 38 | 12 |
| BC_AE | µg m$^{-3}$ | 1.40 | 2.8 | 5.9 | 3.1 | 1.2 |
| BC_PAS | µg m$^{-3}$ | 0.47 | 0.94 | 1.72 | 0.92 | 0.30 |
| EC_RT-TOT | µg m$^{-3}$ | 1.11 | 2.2 | 3.3 | 2.1 | 0.7 |
| EC_TOT | µg m$^{-3}$ | 0.52 | 0.97 | 2.1 | 1.09 | 0.43 |
| OC_RT-TOT | µg m$^{-3}$ | 2.0 | 3.7 | 6.8 | 3.8 | 1.4 |
| OC_TOT | µg m$^{-3}$ | 2.8 | 4.9 | 10.2 | 5.4 | 1.9 |
| Levoglucosan | ng m$^{-3}$ | 129 | 393 | 717 | 387 | 153 |
| Mannosan | ng m$^{-3}$ | 9.0 | 25 | 58 | 28 | 14 |
| Galactosan | ng m$^{-3}$ | 4.3 | 16.1 | 33 | 16.0 | 7.2 |
| $\Sigma$MAs | ng m$^{-3}$ | 143 | 443 | 807 | 431 | 173 |
| Arabitol | ng m$^{-3}$ | 3.2 | 6.5 | 19.3 | 7.5 | 3.9 |
| Mannitol | ng m$^{-3}$ | 1.56 | 3.4 | 19.9 | 4.7 | 3.9 |
| $N \times 10^{-3}$ | cm$^{-3}$ | 4.1 | 8.9 | 17.1 | 9.3 | 3.2 |
| $UF \times 10^{-3}$ | cm$^{-3}$ | 2.9 | 6.4 | 14.5 | 6.9 | 2.6 |
| $SO_2$ | µg m$^{-3}$ | 0.40 | 6.2 | 20 | 8.1 | 4.3 |
| $O_3$ | µg m$^{-3}$ | 2.3 | 15.8 | 58 | 18.4 | 13.7 |
| $NO_x$ | µg m$^{-3}$ | 19 | 83 | 474 | 96 | 62 |
| NO | µg m$^{-3}$ | 2.4 | 23 | 222 | 31 | 30 |
| $CO_2$ | ppm(V) | 446 | 456 | 485 | 456 | 8 |
| $T_{out}$ | °C | 5.4 | 8.4 | 13.7 | 9.0 | 2.3 |
| $T_{in}$ | °C | 18 | 20 | 24 | 20 | 1.4 |
| $RH_{out}$ | % | 38 | 77 | 99 | 77 | 14 |
| $RH_{in}$ | % | 17 | 36 | 46 | 35 | 5 |
| WS | m s$^{-1}$ | 0.7 | 1.6 | 3.1 | 1.6 | 0.6 |

**3.2 Temporal variability**

The time series of $PM_{2.5}$ mass, EC_TOT, OC_TOT, BC_PAS and LVG are shown in Fig. 1 as example. It was concluded previously that a direct coupling between the atmospheric concentration levels and the emission sources, mainly vehicular road traffic, can be identified in central Budapest; nevertheless, the local meteorology and partially long-range transport of air masses have much more influence on the air quality than changes in the source intensity (Salma et al., 2004). This could be
reflected in the concentration variability in Fig. 1, and hence the correlations between the atmospheric concentrations could also be influenced by common effects of the local meteorology for shorter time intervals. Nevertheless, the calm weather





situations usually present during the campaign limited this effect. The pairwise correlation between EC and BC was significant (cf. Table 1). In addition, there were close linear relationships between the $PM_{10}$ mass and $PM_{2.5}$ mass ($r$=0.864), between OC and LVG ($r$=0.809) and between the $PM_{2.5}$ mass and LVG ($r$=0.807). This could indicate that LVG made up a substantial part of the OC. The correlation coefficient between the $PM_{2.5}$ mass and OC was somewhat lower ($r$=0.749), which can imply

differences in their major sources. No direct links between the $PM_{2.5}$ mass and EC, between OC and EC, between EC and LVG, and between $N$ on one side and the $PM_{10}$ mass, $PM_{2.5}$ mass, OC, EC, LVG on the other side were obtained. These suggest that the major sources of EC are different from BB. At the same time, soot particles contribute only partially to the total particle number, which points to an additional important source of particles even in the city centre. This can be atmospheric nucleation (Salma et al., 2014). It is also noted that the correlation coefficients between LVG on one side and MAN ($r$=0.925) and GAN

($r$=0.965) on the other side were large, while the correlations with ARL and MAL were small ($r$=0.629 and 0.204, respectively). The two sugar alcohols did not correlate ($r$=0.576), which suggests that at least one of them had an additional substantial emission source than fungi. This is consistent with an earlier observation according to which ARL and MAL were found to be highly correlated throughout the year, except for winter (Burshtein et al., 2011). Humidity was previously found to be a factor effecting the fungal activity (Cox and Wathes, 1995). It was observed that fungi are more abundant when the RHs are high in

both indoor and ambient air. We could not identify significant correlations in our data set between the RH on one side and ARL and MAL on the other side, which seems to be an attribute of the winter season.

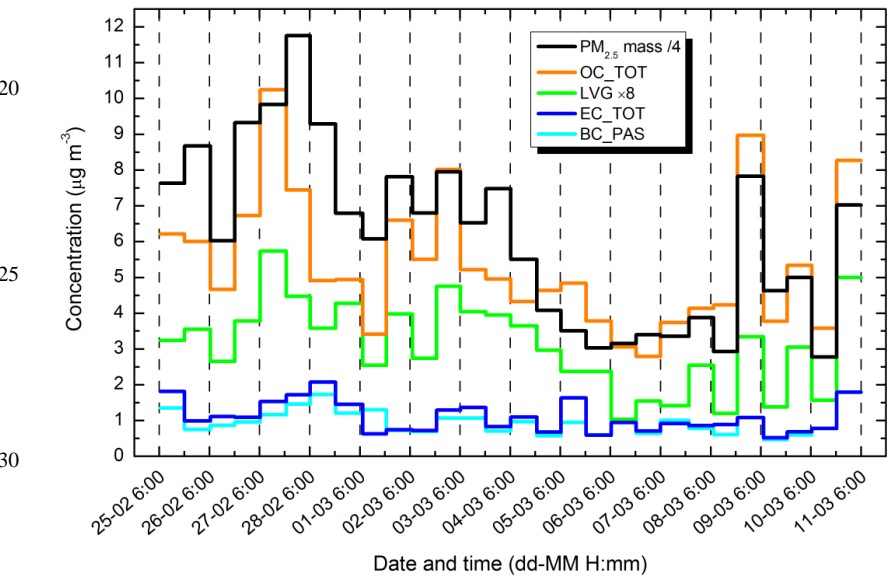



**Figure 1.** Temporal variability of the median atmospheric concentrations for the $PM_{2.5}$ mass, EC and OC determined by the laboratory
OC/EC TOT method (EC_TOT, OC_TOT, respectively), BC derived by PAS, and levoglucosan (LVG) for the sample collection periods of the 2×14 daylight times and nights.

### 3.3 Contributions

On average, EC accounted for 4.8±2.1% of the $PM_{2.5}$ mass. This is smaller than previously observed (14±6%) in a street
canyon in central Budapest in spring, and it is larger than for the near-city background (2.1±0.5%; Salma et al., 2004; Maenhaut et al., 2005). Organic matter made up the $PM_{2.5}$ mass from 21% to 58% with a mean and SD of 37±10%. The mean contribution of EC to TC with its SD was 17.1±4.9%, and the OC/EC concentration ratio varied from 2.4 to 8.9 with a mean and SD of




5.3±1.7%. This means that the carbonaceous matter accounted for 42±11% of the PM$_{2.5}$ mass. The relatively large EC/TC ratio is typical for urban impacts (Salma et al., 2004).

Levoglucosan was utilised to estimate the amount of the PM mass and OC originating from BB. Several PM$_{10}$ mass/LVG and
OC/LVG conversion factors have been used in the literature; they were overviewed by Puxbaum et al. (2007). The conversion factor depends on the burning conditions and wood types. We adopted the factors of 10.7 for the PM$_{10}$ mass from BB, and of 5.59 for the PM$_{2.5}$-fraction OC from BB (OC$_{BB}$), which were suggested by Schmidl et al. (2008) for the mix of wood used in Austria. It was implicitly assumed that the amount of LVG in the coarse size fraction was negligible. This is a reasonable assumption since burning products are predominantly contained in fine particles. The uncertainty of the conversion was
estimated to be approximately 30%. The atmospheric concentration of PM$_{10}$ mass originating from BB varied from 1.4 to 7.7 µg m$^{-3}$ with a median of 4.2 µg m$^{-3}$. The mean contribution of BB to the PM$_{10}$ mass with SD was 11.1±3.4%. The contribution of OC$_{BB}$ to the OC in the PM$_{2.5}$ size fraction ranged from 20% to 60% with a mean and SD of 40±11% (Fig. 2). It can be concluded that BB represents a major source for PM$_{2.5}$ OC and a non-negligible source for the PM$_{10}$ mass. It is mentioned for completeness that the correlation coefficient between LVG and $T_{out}$ was $r=-0.677$. The weather was unusually mild in the
Budapest area during the actual aerosol campaign, so the ordinary BB contributions in winter are expected to be larger than this value. The present results and conclusions on BB are in line with other data from the Carpathian Basin and with various other locations in European cities (Caseiro et al., 2009; Piazzalunga et al., 2011; Maenhaut et al., 2012a). Puxbaum et al. (2007) reported a BB contribution to OM of 28% for the K-puszta station, which represents a rural background or regional site in the Carpathian Basin. As far as the measured monosaccharide anhydrides and sugar alcohols are concerned, their joint contribution
to the OC was determined from their molecular formula, which resulted in a mean and SD of 3.7±0.9%, and the corresponding values for ARL were 0.07±0.03% (see below).

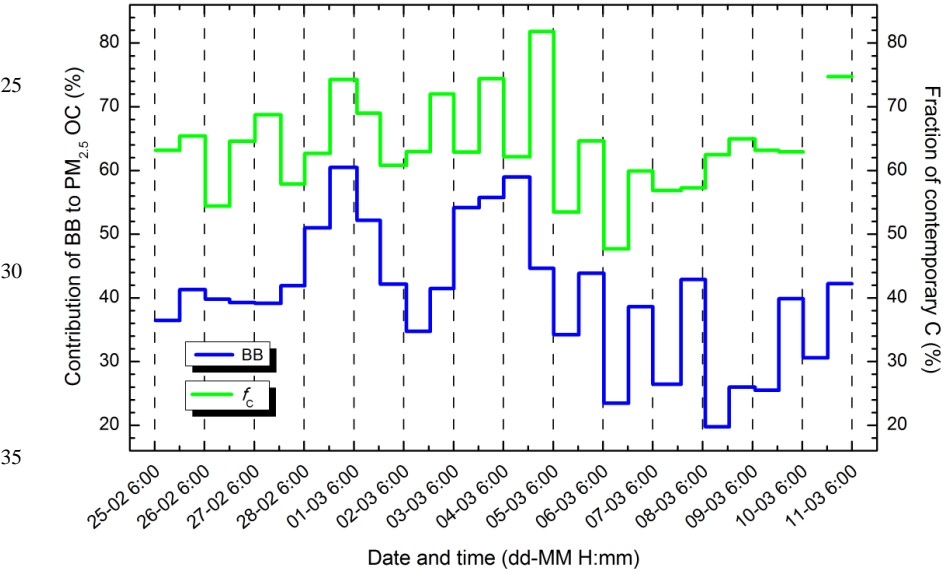

**Figure 2.** Temporal variability of the relative contribution of BB to the PM$_{2.5}$-fraction OC assessed by the levoglucosan marker method, and
of the fraction of contemporary carbon ($f_C$) derived by the radiocarbon marker method.

The concentration ratio LVG/(MAN+GAN) was proposed to differentiate between wood burning and other BB emissions, while the ratio LVG/MAN was applied to distinguish between hardwood and softwood burning emissions (Fine et al., 2004; Schmidl et al., 2008; Fabbri et al., 2009; Caseiro et al., 2009; Favez et al., 2010; Piazzalunga et al., 2011; Maenhaut et al.,





2012a). The typical ranges of the two ratios for different BB sources were overviewed by Maenhaut et al. (2012a). Softwood combustion typically yields a LVG/MAN ratio <4, while the same ratio for hardwood emissions is 14–15. Emissions from lignite and peat burnings result in ratios of 54 and 8.6, respectively. Derivatives of crude oil, natural gas, coal and biomass are the major carbonaceous fuels utilised in Hungary; peat is not burned. The major form, 88% of the consumption expressed in

tons of solid fuels utilised, is lignite/brown coal. For our samples, the LVG/(MAN+GAN) ratios ranged from 6.3 to 11.0 with a mean and SD of 9.2±1.2, while the LVG/MAN ratio varied from 9.8 to 18.5 with a mean and SD of 14.6±2.4. The mean values are at the higher end of the intervals of the ordinary ratios, and may indicate that lignite burning provides a non-negligible contribution to the monosaccharide anhydrides. Nevertheless, the mean LVG/MAN ratio was further utilised formally to calculate the share of softwood (ordinary spruce) burnt relative to the total amount hardwood burnt according to

the relationship of %spruce=(14.8−LVG/MAN)/0.112 derived by Schmidl et al. (2008). The dependency was originally derived for the combustion of common hardwood (beech and oak) and softwood species (spruce and larch) in wood stoves in Austria. The individual LVG/MAN ratios were larger than 14.8 in half of our samples (in 14 cases), and the rest yielded a mean %spruce and SD of 20±13%. The relative uncertainty of the mean is rather large (69%). Moreover, considering the effect of the excluded data, the overall contribution of softwood burning would be even smaller, which seems too low for Hungary,

and points again to the possible importance of lignite burning in the area. It is worth mentioning that the calculation also raises the question whether the whole approach is justifiable for other regions or countries than originally considered.

The C isotope analysis was performed on the TC content of both the front and back quartz filters. The uncorrected back/front filter ratio for TC determined by using the laboratory OC/EC analyser in the TOT mode varied from 12% to 51% with a mean

and SD of 29±10%, and the field blank filters contained 8.9±3.4 µg TC in general. These indicate the importance of using the tandem filter correction method for TC (and OC) in aerosol samples collected by low-volume collection devices. These data also raise the issue whether the back filters which contain adsorbed volatile organic compounds (VOCs) exhibit identical $^{14}C/^{12}C$ isotope ratio as the carbonaceous aerosol particle on the front filters. To investigate this, all back filters were analysed by the AMS method. The range and mean fraction of contemporary carbon with SD for the front filters were 59–83%, 70±7%,

respectively, while the same properties for the back filters were 28–122%, 75±24%, respectively. There was one individual values above 110%, which can be explained only by some anthropogenic $^{14}C$ sources, such as medical or other industrial release. As a conclusion, the $f_C$ values of the back filters were individually taken into account for the front filters. Its extent resulted in a range, mean back-to-front ratio and SD of 10–49%, 25±10%, respectively. These imply that the tandem filter correction becomes necessary for the radiocarbon method on low-volume samples (if the TC is less than 1 mg on a filter). The

situation can be different for high-volume samplers. The corrected $f_C$ is also shown in Fig. 2 as time series. The contribution of contemporary C to the TC varied from 48% to 82% with a mean and SD of 64±7%. Radiocarbon data were obtained for Budapest for the first time. Excitingly, the correlation coefficient between LVG and $f_M$ was modest, $r$=0.523, which suggests that the contribution of OC from the other possible major source of modern carbon, i.e. from biogenic sources was substantial. Backward air mass trajectories showed that the fossil impact was larger for local air masses (sources), while an enhanced non-

fossil fraction was generally observed for long-range transported air masses.

### 3.4 Coupled source apportionment

The relative contributions of EC and OC to the TC derived directly from the measured atmospheric concentrations were combined with the results of the independent radiocarbon and LVG marker models regarding the fossil, contemporary (non-fossil) and BB sources in a coupled approach (cf. Bonvalot et al., 2016) on a sample by sample basis. The novel source

apportionment scheme of the TC into the contributions of EC and OC from FF combustion (EC$_{FF}$ and OC$_{FF}$, respectively), EC and OC from BB (EC$_{BB}$ and OC$_{BB}$, respectively), and OC from biogenic sources (OC$_{BIO}$) proposed and utilised in the present study is summarised in Fig. 3. It consists of pragmatic and effective attribution steps, which are expressed by multiplication





factors. The factor $f_1$ was set to $f_1=f_C$ of the actual sample. The modest correlation between LVG and EC (see Sect. 3.2) revealed that BB alone represents relatively a less substantial source of EC relative to TC than the joint contributions of FF combustion and BB. For this reason, the relative contribution of $EC_{BB}$ was estimated by adopting the mean EC/OC values previously reported explicitly for BB. Szidat et al. (2006 and references therein) utilised a critically evaluated ratio of $(EC/OC)_{BB}=16\pm5\%$,

and Bernardoni et al. (2013) derived a ratio and SD of $18\pm4\%$ for wood burning. Their mean value of $(EC/OC)_{BB}=17\%$ was utilised in the present calculation scheme. The factor $f_2$ was determined from the mass balance equation: $f_1\times f_2\times TC/(5.59\times LVG)=(EC/OC)_{BB}$, which yielded the multiplication factor $f_2=5.59\times LVG\times(EC/OC)_{BB}/f_1/TC$. The LVG and TC data refer to the measured atmospheric concentrations for the actual sample, while the $OC_{BB}$/LVG ratio of 5.59 is discussed in Sect. 3.3 and also below. The relative contribution of BB to the contemporary OC was assessed by a multiplication factor

$f_3=5.59\times LVG/f_1/(1-f_2)/TC$, which was obtained from the mass balance equation: $f_1\times(1-f_2)\times f_3\times TC=5.59\times LVG$. The remaining fraction of $(1-f_3)$ of the contemporary OC was considered as the relative contribution from biogenic sources. The TC from fossil sources was divided into the relative contribution of $EC_{FF}$ in that way so the weighted joint contributions of the EC from the FF combustion and BB become equal to the actual EC/TC ratio for the given sample, thus from the mass balance equation: $f_1\times f_2+(1-f_1)\times f_4=EC/TC$, which yielded the multiplication factor $f_4=(EC/TC-f_1\times f_2)/(1-f_1)$. The main advantage of this

apportionment method is its straightforward character and the fact that the required data are usually available in similar studies, while its main limitation is in the bias between the multiplication factors $f_2$ and $f_3$. The mean factors and their SDs averaged for all samples are $f_1=64\pm7\%$, $f_2=9.0\pm1.9\%$, $f_3=58\pm14\%$, $f_4=31\pm11\%$.

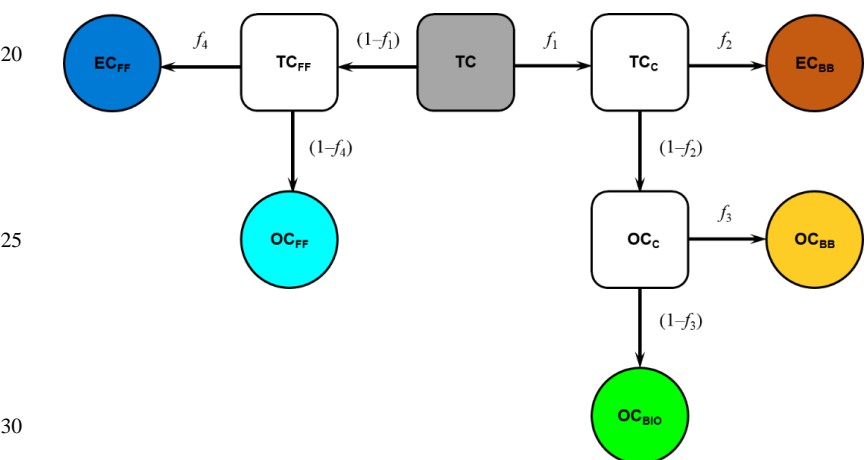



**Figure 3.** Source apportionment scheme based on the coupled radiocarbon-levoglucosan marker method for the relative contributions of EC and OC from fossil fuel (FF) combustion, biomass burning (BB) and biogenic sources (BIO) to the total carbon (TC=EC+OC). The multiplication factors are explained in the text. The subscript C indicates contemporary carbonaceous fractions.


The mean relative contributions of carbonaceous species to the TC with SDs derived by averaging for all samples were $11.0\pm4.2\%$ for the $EC_{FF}$, $25\pm6\%$ for the $OC_{FF}$, $5.8\pm1.4\%$ for the $EC_{BB}$, $34\pm8\%$ for the $OC_{BB}$, and $24\pm9\%$ for the $OC_{BIO}$. The latter contribution also includes the mean share of the primary organic aerosol emitted by fungi of ca. 0.02%, which was assessed by using ARL (see Sect. 3.5 and Fig. 8). The relative contribution of fungal spores is rather small, which expectedly

remains so for the $PM_{10}$ size fraction as well, but it can have biological relevance due to its possible allergenic influence. Biomass burning and FF combustion sources contributed similarly by $40\pm10\%$ and $36\pm7\%$, respectively, while biogenic sources made up $24\pm9\%$ of the TC concentration. The median relative contributions are shown in Fig. 4. It has to be mentioned that the $(OC/LVG)_{BB}$ conversion factor is based on laboratory studies which mainly considered primary particles (emission products), although SOA from wood burning can yield amounts that are comparable to primary particles (Szidat et al., 2009).





This can cause underestimation of the $OC_{BB}$ contributions, and consequently, overestimation of the $OC_{BIO}$ contributions. Further important uncertainty can arise from a variable OC/LVG conversion factor of 5.59 due to spatially and temporally changing burning conditions. The overall relative contributions are in very good agreement with other wintertime urban atmospheric studies (Szidat et al., 2009 and references therein; Minguillón et al., 2011; Bernardoni et al., 2013; Bonvalot et

al., 2016), and the estimate of the $OC_{BIO}$ contribution during the winter season is also consistent with model calculations (Simpson et al., 2007).

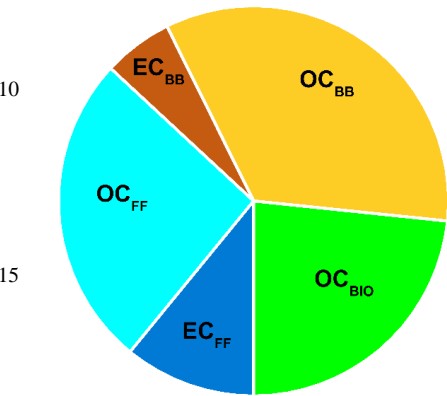

**Figure 4.** Median relative contributions of $EC_{FF}$ (11.2%), $OC_{FF}$ (27%), $EC_{BB}$ (5.9%), $OC_{BB}$ (34%) and $OC_{BIO}$ (24%) to the total carbon with
a median atmospheric concentration of 6.0 μg m$^{-3}$ in the PM$_{2.5}$ size fraction in central Budapest.

The individual relative contributions of the carbonaceous species to the TC were converted to their share in the PM$_{2.5}$ mass as well. An OM/OC conversion factor of 1.6 was adopted in the calculations (see Sect. 2.2). The mean relative contributions to the PM$_{2.5}$ mass with SDs derived by averaging for all samples were 3.1±1.6% for the $EC_{FF}$, 11.1±4.3% for the $OM_{FF}$,
1.53±0.40% for the $EC_{BB}$, 14.4±3.8% for the $OM_{BB}$ and 11.1±6.1% for the $OM_{BIO}$. The importance of BB sources, FF combustion and biogenic sources for the PM$_{2.5}$ mass was largely similar to each other, namely approximately 15%, 14% and 10%, respectively. We are aware that the conversion factor can change for organic species from different source types, and therefore, the latter (PM$_{2.5}$ mass) apportionment should be considered with caution. Both sets of the relative contributions are expected also to change substantially for various seasons or on an annual scale due to important changes or time patterns in
heating, other human activities, formation pathways and biogenic emission strengths.

### 3.5 Apportioned carbonaceous species

The atmospheric concentrations of the apportioned carbonaceous chemical species and TC are shown in Table 3.

**Table 3.** Range, median, mean with standard deviation (SD) of the atmospheric concentrations for the apportioned EC and OC from FF combustion ($EC_{FF}$ and $OC_{FF}$, respectively), EC and OC from BB ($EC_{BB}$ and $OC_{BB}$, respectively), OC from biogenic sources ($OC_{BIO}$) and for
the measured TC in μg m$^{-3}$ for the PM$_{2.5}$ size fraction.

| Species | Min | Median | Max | Mean | SD |
|---------|-----|--------|-----|------|-----|
| $EC_{FF}$ | 0.31 | 0.68 | 1.43 | 0.69 | 0.29 |
| $OC_{FF}$ | 0.53 | 1.52 | 2.8 | 1.60 | 0.59 |
| $EC_{BB}$ | 0.122 | 0.38 | 0.68 | 0.37 | 0.14 |
| $OC_{BB}$ | 0.72 | 2.3 | 4.0 | 2.2 | 0.8 |
| $OC_{BIO}$ | 0.38 | 1.30 | 3.4 | 1.60 | 0.85 |
| TC | 3.5 | 6.0 | 11.8 | 6.5 | 2.1 |





The properties and relationships among the apportioned carbonaceous species were investigated by pairwise correlations. Selected scatter plots are shown in Figs. 5–8. It is seen that there was no meaningful linear relationship between $EC_{FF}$ and $EC_{BB}$ ($r=0.340$, Fig. 5 upper panel) and between $EC_{FF}$ and $OC_{FF}$ ($r=0.170$, Fig. 7 upper panel). All three apportioned OC

species showed tendentious links with each other. The correlation coefficient between $OC_{FF}$ and $OC_{BB}$ was $r=0.458$ (Fig. 5 lower panel), and they were $r=0.431$ and $0.432$ between $OC_{BIO}$ on one side and $OC_{FF}$ and $OC_{BB}$ on the other side, respectively (Fig. 6). This suggests that the formation processes of OC species from anthropogenic VOCs and biogenic VOCs (BVOCs) were primarily influenced or controlled by a common factor, which is most likely the atmospheric photochemistry. This effect is, however, expected to be realised in a complex way since the relationship of GRad on one side with the three apportioned

OC species on the other side showed only fluctuations. As far as the air temperature is concerned, the dependency of $OC_{BB}$ on $T$ was only tendentious ($r=-0.661$, Fig. 7 lower panel). Emission of BVOCs (e.g. monoterpenes) can be described by an exponential $T$ dependency, thus $OC_{BIO} \propto BVOC \propto \exp(a \times T)$, where $a$ is a constant (Kontkanen et al., 2016). Nevertheless, we could not identify any obvious link between $T$ and $\log(OC_{BIO})$ or $OC_{BIO}$ probably because of the narrow $T$ range during the campaign, and because the transformation of BVOCs from gaseous phase to aerosol phase takes place in a complex system

depending sensitively on many other multifactorial chemical and atmospheric conditions, which are not expressed obviously by pairwise correlations. The moderate pairwise correlations between the apportioned OC species also point to the relevance and role of primary organic matter (POM) from FF combustion, BB and biogenic sources. At the same time, there was a tendentious and strong linear relationship between $NO_x$ – which is emitted by 60–70% from road vehicles in Budapest – and $EC_{FF}$ ($r=0.823$, Fig. 8 upper panel), while the correlation coefficient between $OC_{FF}$ and $NO_x$ was not significant ($r=0.037$).

Arabitol, which expresses the primary emissions from fungi, and which possibly can be also related to somewhat more general biogenic activity showed a tendentious dependency on the $OC_{BIO}$ (Fig. 8 lower panel). By excluding the two data points (1D and 1N), which appear outliers, a correlation coefficient $r=0.494$ was obtained. It has to be noted that primary biological emissions (including ARL) are mainly associated with the coarse size mode, while the $PM_{2.5}$ size fraction investigated in the present study overlaps only partially with it. Stronger links between ARL and $OC_{BIO}$ are expected to be obtained by considering

ARL in coarse particles.

These results altogether can be interpreted by concluding that 1) there were various substantial FF combustion sources active in the area which result in different EC/OC ratios, 2) $EC_{FF}$ was mainly emitted by vehicular road traffic, 3) the contribution of non-vehicular fossil sources such as domestic and industrial heating or cooking using gas, oil or coal to $OC_{FF}$ was substantial,

4) the mean contribution of BB to soot particles was smaller by a factor of approximately 2 than of road traffic, and 5) formation of OC from fossil, BB and biogenic VOCs were jointly influenced by photochemistry, while POM from these sources also played a relevant role.



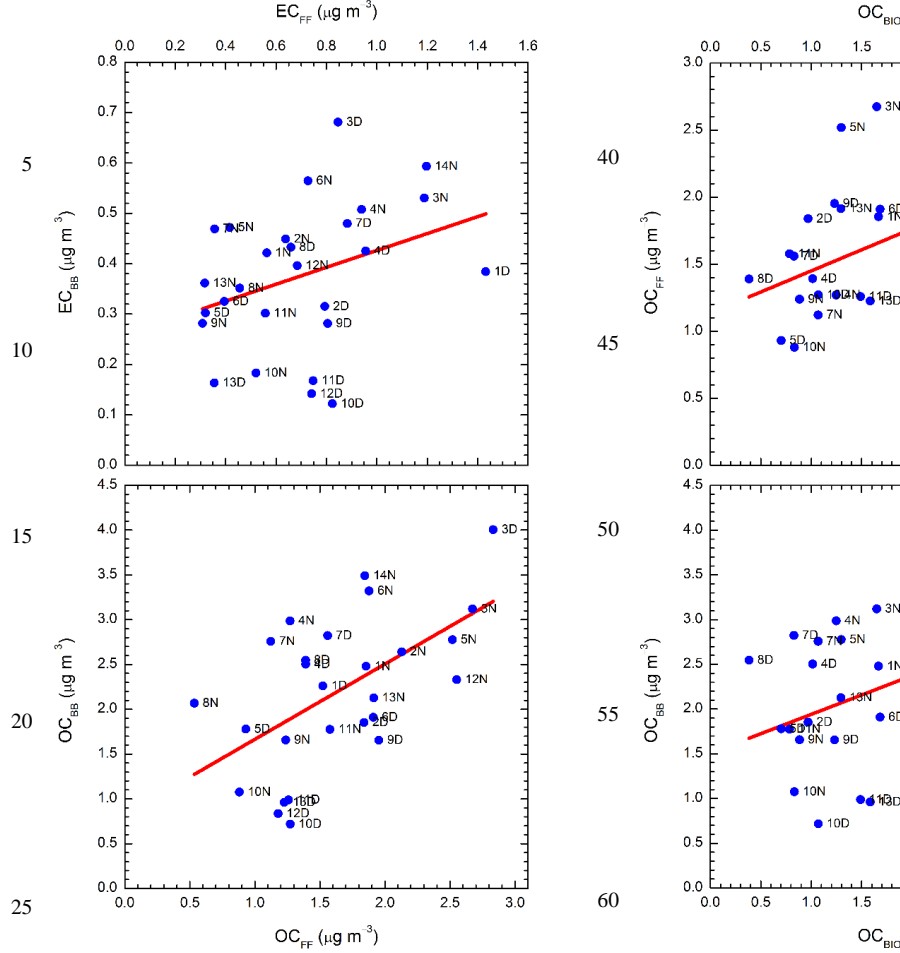

**Figure 5.** Scatter plots between apportioned atmospheric concentrations of $EC_{FF}$ and $EC_{BB}$, and of $OC_{FF}$ and $OC_{BB}$ for the $PM_{2.5}$ size fraction in central Budapest. The red lines represent a linear fit to the data. The order number of the samples together with daylight time (D) or night (N) periods are indicated by labels next to the data points.

**Figure 6.** Scatter plots between apportioned atmospheric concentrations of $OC_{BIO}$ and $OC_{FF}$, and of $OC_{BIO}$ and $OC_{BB}$, for the $PM_{2.5}$ size fraction in central Budapest. The red lines represent a linear fit to the data. The order number of the samples together with daylight time (D) or night (N) periods are indicated by labels next to the data points.



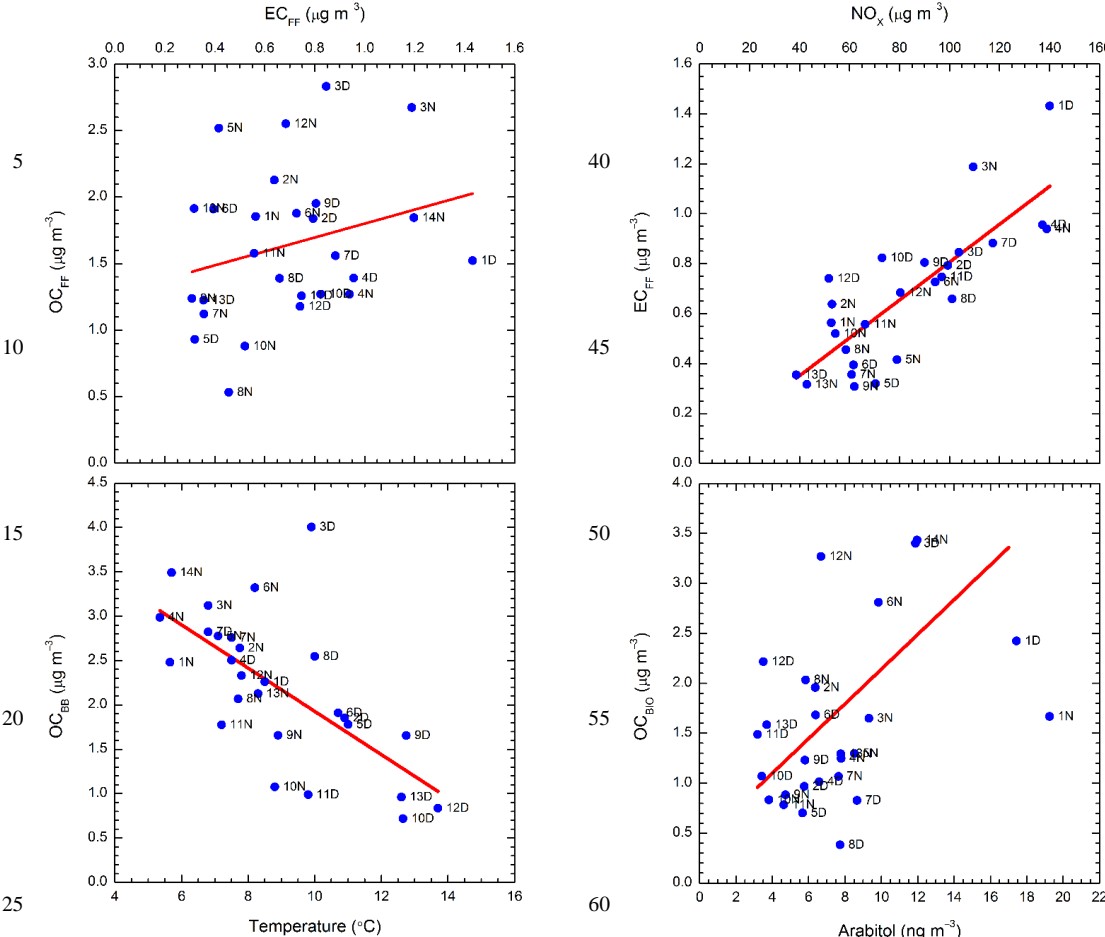

**Figure 7.** Scatter plots between apportioned atmospheric concentrations of $EC_{FF}$ and $OC_{BB}$, and of $OC_{BB}$ and air temperature ($T$) for the $PM_{2.5}$ size fraction in central Budapest. The red lines represent a linear fit to the data. The order number of the samples together with daylight time (D) or night (N) periods are indicated by labels next to the data points.

**Figure 8.** Scatter plots between atmospheric concentrations of $NO_x$ and apportioned $EC_{FF}$, and of arabitol and apportioned $OC_{BIO}$ for the $PM_{2.5}$ size fraction in central Budapest. The red lines represent a linear fit to the data. The data points 1D and 1N on the lower panel were excluded from the data set when fitting. The order number of the samples together with daylight time (D) or night (N) periods are indicated by labels next to the data points.

**4 Conclusions**

We showed here that BB was responsible for 40% of the carbonaceous chemical species in the $PM_{2.5}$ size fraction in central Budapest during a mild winter without no snow cover in the larger area, while FF combustion contributed by 37%, and biogenic sources made up 24%. $EC_{FF}$ and $OC_{FF}$ were associated with different FF combustion sources. Most emission of the former species was caused by road traffic, in particular diesel driven vehicles, while most $OC_{FF}$ was attributed to other fossil source types. The main formation process of all three OC species (i.e. $OC_{FF}$, $OC_{BB}$ and $OC_{BIO}$) from anthropogenic VOCs and BVOCs were influenced by a common factor, which is most likely the atmospheric photochemistry. This effect was, however, realised in a complex multifactorial way, and the role of POM was also important. The relative contribution of BB to the $PM_{10}$ mass concentration was modest, approximately 11%. The corresponding contributions are usually larger in many Western and Northern European cities. Our value seemingly indicates limited possibilities for implementing action plans for air quality



improvements by controlling BB. Nevertheless, reducing soot and VOC emissions from BB could result in a substantial decrease up to about 40% of organics in the PM$_{2.5}$ size fraction. This chemical fraction and particle size range do contain most of the potentially harmful, toxics and reactive organic compounds (e.g. polyaromatic hydrocarbons), intermediates or other particulate products from burning stained or processed wood. In addition, all and the most severe daily PM$_{10}$ health limit

exceedances in Budapest have occurred in winter when the contribution of BB is expected to be the largest, and when the BB takes place in many individual residences in the region during the same time interval, e.g. under cold weather conditions. Technological improvements and control measures for various (mostly household) appliances that burn biomass and wood, together with efficient education and training of their users, in particular on the admissible fuel types offer rather important potentials for improving the air quality in Budapest, and represent an important form of societal implications of atmospheric

aerosols in cities in general.

Further improvements in the source apportionment can be achieved by utilising the coupled radiocarbon-levoglucosan marker method on corresponding sets of different carbonaceous chemical fractions such as OC, EC, water-soluble OC and water-insoluble OC. This method possesses further potentials to supply more detailed results and important information on emission

and formation processes of carbonaceous chemical species. In this perspective research on a yearly time scale, the sample collection and analytical protocols need to be optimised jointly, and the conclusions reached in the present study are to serve as basis for these dedicated plans.

## 5 Data availability

The observational data used in this paper are available at http://salma.web.elte.hu/BpArt of the Eötvös University, Hungary,

or on request from the corresponding author.

**Acknowledgements.** The authors are grateful to G. Močnik of Aerosol d.o.o., Slovenia for providing the Aethalometer for the measurements, and A. Kardos of the Eötvös University for his help in the aerosol campaign. Financial supports by the Hungarian Scientific Research Fund (K116788), by the Széchenyi 2020 programme, the European Regional Development Fund and the Hungarian Government (GINOP-2.3.2-15-2016-00028), and by the European Union and the State of Hungary,
co-financed by the European Regional Development Fund (GINOP-2.3.2.-15-2016-00009 'ICER') are also appreciated. W.M. and M.C. are indebted to R. Vermeylen for assistance in the GC/MS analyses and to the Belgian Federal Science Policy Office for financial support.

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
