# Peer review of "Source apportionment of carbonaceous chemical species to fossil fuel combustion, biomass burning and biogenic emissions by a coupled radiocarbon-levoglucosan marker method"

_Atmospheric Chemistry and Physics, 2017_

## Referee Comment (RC1) · Anonymous Referee #1 · 5 Jul 2017

The paper "Source apportionment of carbonaceous chemical species to fossil fuel combustion, biomass burning and biogenic emissions by a coupled radiocarbon-levoglucosan marker method" by Salma et al. aims at proposing a source apportionment methodology based on simultaneous measurements of 14C in the total carbon (TC) fraction of carbonaceous aerosol and levoglucosan measurements.

The approach can be of interest for the scientific community. The source apportionment approach proposed in the paper, joining 14C measurements on TC and levoglucosan determination, can provide information similar to those obtained by 14C measurements

in OC and EC - which are still performed by few groups in the world - even if further a-priori assumption on (LVG/EC)BB emission ratio is needed. Nevertheless, different criticalities are present in the current version of the manuscript. Thus, the paper needs to be thoroughly revised (major revision) before being considered for publication on ACP.

The main aspects requiring attention are the following:

- source apportionment scheme description should be re-written to better clarify the information flow. Figure 3 has to be completely revised;

- indication on source apportionment uncertainties is mostly missing, some hints for their estimate should be added to the text; furthermore, possible role of secondary organic compounds on OC-BIO source apportionment needs further comments;

- assumptions needed for the application of the approach should be stated more clearly;

- high time resolved measurements of carbonaceous fractions and aerosol absorption/extinction properties were performed in parallel to filter sampling, but they are rarely considered in data analysis. I suggest the authors to consider one of the following possibilities: a) completely remove the description of techniques and the introduction on paragraph 3; b) revise optical properties data (currently, data treatment is not adequate, see detailed comments) and exploit multi-wavelength measurements to attempt optical source apportionment approaches, i.e. Aethalometer model (Sandradewi et al., 2008 Environ. Sci. Technol. 42, 3316-3323 and following modifications) or Multi-Wavelength Absorption Analyser model (Massabò et al., 2015. Atm. Environ, 108, 1-12). These models provide information on fossil fuel combustion/biomass burning contribution to carbonaceous aerosol basing on optical absorption measurements and thermal-optical OC/EC/TC. The information could support the results obtained by the 14C on TC and LVG source apportionment proposed in the paper.

Major comments

- page 4, par. 2.2: indication on minimum detection limits and uncertainty for the mentioned techniques are completely missing. Please, add such information (a table should be enough)

- page 5, line 37. The authors used the same fm reference for modern material (1.08), independently on the source (BB or BIO). Usually in 14C source apportionment studies, two different values are chosen for BB (expected higher due to integrated signal) and BIO (which is expected to be in equilibrium with today atmospheric 14C/12C ratio. The approximation should be highlighted (see e.g. par 2.4 Minguillòn et al., 2011 cited by the authors)

- page 6, line 2: FDMS usually operates at 30°C. Any comment about possible mass losses (e.g. nitrate?)
* * *
The following two comments are of interest only if the authors decide not to remove the part concerning high time resolved optical and thermal-optical measurements

- page 6, line 6 and 9-10. Basing on Drinovec et al. (2015), Atmos. Meas. Tech., 8, 1965–1979 (page 1970) I guess that 16.6 m2/g is the mass extinction (not absorption) coefficient: Aethalometer assumes proportionality between babs and batn through a proportionality factor C - Weingartner et al, (2003), J. Aerosol Sci., 34, 1445–1463. The mass absorption coefficient historically considered by Aethalometer @880 nm is 7.77m2/g (Drinovec et al. (2015)). If a-priori assumption has to be applied to convert PAS measurements into BC concentration maintaining equivalence with what performed in the Aethalometer, then 7.77m2/g is the value to be extrapolated to 1064 nm for application to PAS measurements (please evaluate also the following comment).

- page 6, lines 18-23: the comparisons between optical EBC (from AE or PAS) and thermal EC are strongly affected by the assumption on absorption/extinction coefficients

chosen a-priori. As such values are site/season/composition dependent, the parallel availability of optical (batn/babs) and thermal (EC) measurements could be better exploited to provide information on BC mass absorption coefficients in Budapest;
* * *
- page 7, line 37-38: in the cited works, SFUs are reported to have a cut-off diameter of $2\mu$m. Did the authors change e.g. nuclepore filter pore size or flow-rate to modify the size cut? Different cut-off size could partially justify differences, as well as the use of different instrumentation (i.e. SFU vs. TEOM) or the improvement of combustion technologies in the last 15 years, leading to smaller particles formation.

- page 10, lines 9-15. Lignite contribution and its high LVG/MAN does not allow reliabile softwood/hardwood quantification. I suggest to completely remove this paragraph.

- page 11, line 27: "the fC values of the back filters were individually taken into account for the front filters". Please better explain the procedure for front filter correction.

- page 12, line 5: from Bernardoni et al. (2013) I guess the mentioned value was obtained elsewhere (Bernardoni et al, 2011, Sci Total Environ, 409, 4788–4795)

- page 12, figure 3 and lines 9-14: I don't think the chart correctly represents information flow described in the text. Chart information flow makes argue that factor f2 and f3 are needed to determine EC-BB and OC-BB from TC-C and OC-C. This interpretation would be nonsense, as no proportionality is expected between EC-BB and TC-C, as well as between OC-BB and OC-C (due to at least an independent source - BIO - contributing to OC-C). Nevertheless, the text shows that the source apportionment follows a completely different path. If I correctly interpreted the text, it should be evidenced in the chart that input data are obtained from 14C information (TC-C), and LVG (OC-BB and EC-BB thanks to emission ratios). f2 and f3 are then a by-product (by ratios or subtractions) of these quantities and do not have anything to do with "mass balance equations", opposite to what reported at line 10. Then, I guess from the text that EC-FF

is derived as difference between total EC and EC-BB. EC-FF is then exploited to obtain the factor f4. I suggest to completely re-think the chart, including external inputs and their role in the source apportionment scheme. Arrows should start from input data and point to calculated quantities. Furthermore, corresponding text (lines 9-14, page 12) is quite obscure in its current form, and it should be re-written to better evidence the real information flow.

- page 12, lines 14-17: I suggest to mention also a disadvantage of the proposed model compared to the direct determination of 14C in OC and EC fraction, mainly impacting on EC source apportionment: in the case of the proposed model, EC source apportionment requires assumption on wood burning LVG/EC emission ratio, whereas if 14C is directly measured in the EC fraction, the apportionment is straightforward.

- Page 13, line 25: "The importance of BB sources, FF combustion and biogenic sources for the PM2.5 mass". Please note that the model does not apportion the total contribution of these sources to PM2.5, but only the contribution of carbonaceous species emitted by these sources (e.g. high emissions of K+ from BB combustion are expected to impact on PM2.5, but they are - of course - completely neglected by this source apportionment approach). Their total relative impact to PM2.5 is for sure higher, and the proportions among them can be different.

- Page 13, line 31: please, add some considerations on estimated uncertainties related to the model. Page 14, line 16: please add considerations on the possible impact of secondary compounds related to BB emission on OC-BIO estimates

- Page 14, line 19 and lines 26-27: at line 19 the authors find correlations between NOx (assumed as markers for traffic emissions) and EC-FF, whereas no significant correlation is found between NOx and OC-FF. I think this observation supports considerations 2 and 3 at lines 26, 27. Nevertheless, an important role can be played also by secondary OC-FF, justifying disentanglement between primary species from FF combustion and OC-FF. It should also be noticed that to avoid effects due to differ-

ent repartition of NO vs. NO2, NOx concentration should be better represented in ppb for the evaluation of more robust correlation coefficients.

Minor comments - page 3, line 39: what is TEOM size-cut? - page 4, line 8-9: don't agree with the sentence. EUSAAR_2 was initially developed for background sites (see end of introduction in Cavalli et al., 2010 Atmos. Chem. Phys., 11, 10193–10203). Its application has effectively been extended to urban sites. Nevertheless, concerns were posed in the literature on its use in urban areas (e.g. Piazzalunga et al., Atmos. Chem. Phys., 11, 10193–10203, 2011) - page 4, line 11: EUSAAR_2 protocol is longer than 15 minutes. Was it somewhat modified? Please, report, effective protocol - page 4, line 36: why did the authors use different thermal protocols for on-line and off-line instrumentation? - page 8, Table 2: I expected mean values for PM10 being the sum of mean values for PM2.5.and PM10-2.5. It is not so. Is it a time-averaging problem? Or can it be due to non-parallel sampling (e.g. missing values in only one of the two samplers were considered in the average?) - page 9, line 3: LVG is not a substantial part of OC (as can be seen in figure 1). The observation can indicate an important contribution of LVG source (BB) also to other OC species. - page 9, figure 1: please plot EC and BC (if maintained) on a secondary axis to better allow appreciating modulations - page 10, line 1: no % is needed in OC/EC ratio at the beginning of the line. Furthermore, I suggest to comment this datum which gives indication of strong secondary organic aerosol formation. - page 11 line 29: the authors relate the need of sampling artefact correction to the total carbon load on a filter ("1 mg"), and not to the mass/area load. Can they justify this?

---

## Referee Comment (RC2) · Anonymous Referee #2 · 20 Jul 2017

The paper describes two weeks measurement aiming to characterisation of carbonaceous aerosol in Budapest using combined approach that utilizes both organic tracers and radiocarbon data for determination of biomass, fossil fuel, and biological origin particulate matter shares. Besides basic statistics of obtained data, the authors present a new updated scheme how to apportion elemental and organic fractions to their source categories without their separation before radiocarbon analysis. Although their approach can be useful in cases when pre-separation of EC and OC before radiocarbon analysis is not available it of course increases uncertainty of the apportionment of these

fractions. Main added uncertainty to the classical method of radiocarbon analysis with the split of EC and OC before the analysis is included in predefined value of ECBB to OCBB ratio. Nevertheless, the authors do not comment uncertainty connected with this value at all. Moreover, although they found that one of the samples was probably contaminated by artificial 14C release, they omitted to discuss a possibility of such influence on the other samples. Finally, although the authors say that their OCBIO is in line with Simpson et al. (2007) the opposite is true due to different definition of fractions. The observed result was 0.17 $\mu$g/m3 of OCBIO in K-puszta for winter (primary biogenic particles), which was roughly 0.2% of TC in PM10 and ca 20% of TC was attributed to non-fossil SOA. Therefore the reviewer suggests to call the last OC fraction same way (non-fossil SOA), also with regard to the text on page 12, line 44. The authors also spend quite a lot of space describing aethalometer and other online instrumentation results but they did not use an opportunity to test their method with aethalometer source apportionment method. Besides these major issues, several other specific comments are summarized below. Therefore major revisions are needed before publishing the paper in ACP.

Page 4, lines 33-34: No blank uncertainty is given for mass on Nuclepore and quartz filters

Page 6, line 1-2: It seems unusual to combine online PM2.5 mass with offline filters (PM10-2.5) to construct PM10, especially when PM2.5 mass from filters is available even from the same filter pack. Moreover, fine fraction from Ghent SFU is rather PM2 than PM2.5 (Hopke et. al 1997) and therefore coarse fraction will be probably PM10-2 instead of PM10-2.5.

Page 6, line 7 and elsewhere: The term "data lines" should be probably "data points"

Page 12, Fig 3.: The scheme is quite unclear, needs improvement.

Page 14, line 5: The term "tendentious" seems not properly used in the text several times (three times on this page), please check.

Reference: Hopke, P. K., Xie, Y., Raunemaa, T., Biegalski, S., Landsberger, S., Maenhaut, W., Artaxo, P. and Cohen, D. 1997. Characterization of the Gent stacked filter unit PM10 sampler. Aerosol Sci. Technol., 27 726-000000–735

---

## Author Comment (AC1) · 23 Sep 2017

**Response to Referee #1**

The authors thank Referee #1 for his/her detailed, extensive, expertise and valuable comments to further improve and clarify the MS. We have considered all recommendations, and made the appropriate alterations. Our specific responses to the comments are as follows.

**Aspects requiring attention**

1) source apportionment scheme description should be re-written to better clarify the information flow. Figure 3 has to be completely revised;

The apportionment scheme is now described more explicitly. We clarify all input data that are necessary for the apportionment and their actual entrance point into the scheme more systematically in both the text and Fig. 3. The arrows in Fig. 3 were originally selected in such way that they indicate the pathways and their major steps in the data treatment flow starting from the measured TC concentration toward the assessed end quantities of the five major carbonaceous species, i.e. $EC_{FF}$ and $OC_{FF}$, $EC_{BB}$ and $OC_{BB}$, and $OC_{BIO}$, which is considered helpful in overviewing the idea of the proposed new approach for readers, who are not very familiar with source apportionments.

2) indication on source apportionment uncertainties is mostly missing, some hints for their estimate should be added to the text; furthermore, possible role of secondary organic compounds on OC-BIO source apportionment needs further comments;

We added now a brief discussion on the model uncertainties and assessed their extent. We also completed the text with a comment on the possible role of secondary organics on $OC_{BIO}$.

3) assumptions needed for the application of the approach should be stated more clearly;

The assumptions are now formulated more explicitly. This was achieved within the modifications indicated in reply to comment Aspects requiring attention 1.

4) high time resolved measurements of carbonaceous fractions and aerosol absorption/extinction properties were performed in parallel to filter sampling, but they are rarely considered in data analysis. I suggest the authors to consider one of the following possibilities: a) completely remove the description of techniques and the introduction on paragraph 3; b) revise optical properties data (currently, data treatment is not adequate, see detailed comments) and exploit multi-wavelength measurements to attempt optical source apportionment approaches, i.e. Aethalometer model (Sandradewi et al., 2008 Environ. Sci. Technol. 42, 3316-3323 and following modifications) or Multi-Wavelength Absorption

Analyser model (Massabò et al., 2015. Atm. Environ, 108, 1-12). These models provide information on fossil fuel combustion/biomass burning contribution to carbonaceous aerosol basing on optical absorption measurements and thermal-optical OC/EC/TC. The information could support the results obtained by the 14C on TC and LVG source apportionment proposed in the paper.

The authors regret that they did not indicate to the handling editor and the potential referees at the submission stage that another MS, which deals with and evaluates the data from the on-line measurements obtained by the AE, PAS and DMPS in focus, is under preparation. This other MS will include the evaluation methods referred to by the Referee, thus the so-called Aethalometer model (the wavelength dependence of the optical absorption coefficient) for both the AE (Sandradewi et al., Environ. Sci. Technol., 42, 3316–3323, 2008; Sandradewi et al., Atmos. Environ., 42, 101–112, 2008; Favez et al., Atmos. Environ., 43, 3640–3644, 2009; Favez et al., Atmos. Chem. Phys., 10, 5295–5314, 2010) and PAS–DMPS (Ajtai et al., Atmos. Environ., 122, 313–320, 2015) data sets. In addition, a new approach of combining the PAS, DMPS (number size distribution) and AE (soot) data sets into one model is also planned. Comparing the results, advantages and limitations of the filter-based approach (in the present paper) and several optical-related source apportionment methods based on the on-line data sets will be one of the key points in this other MS. We decided to split the whole matter into two parts (with a natural division line between the collection of aerosol samples and on-line measurements) because we consider that their joint presentation 1) could be too lengthy and perhaps even too complex and likely could have a too broad scope, and therefore, 2) it would detract the attention from the new apportionment scheme proposed in the present MS, and 3) our goal is not to prepare a comparative MS on a selection of apportionment methods. We indicated now these additional arguments, motivations and aims very briefly in the text. Nevertheless, it seems plausible and advantageous to describe the experiments of the aerosol sample collection and measurement campaign together in order to have a more comprehensive overview on the aerosol campaign and resulting analytical data sets as a whole, and to assist the future MS to focus the attention specifically on modelling issues and data validation.

**Major comments**

1) page 4, par. 2.2: indication on minimum detection limits and uncertainty for the mentioned techniques are completely missing. Please, add such information (a table should be enough)

We included now into the text the determination limits of some measuring techniques where they were missing. We also indicated explicitly that the measured concentrations were at least

several times larger than the determination limits due to the common levels that are present in the city centre and to the well-selected sampling/integration times, which also imply that the atmospheric variability and the uncertainties related to conversions (e.g. from OC and OM, or from LVG to $OC_{BB}$) can cause larger uncertainties that the analytical techniques themselves.

2) page 5, line 37. The authors used the same fm reference for modern material (1.08), independently on the source (BB or BIO). Usually in 14C source apportionment studies, two different values are chosen for BB (expected higher due to integrated signal) and BIO (which is expected to be in equilibrium with today atmospheric 14C/12C ratio. The approximation should be highlighted (see e.g. par 2.4 Minguillòn et al., 2011 cited by the authors)

Minguillòn et al. concluded that the differences in the $f_C$ caused by the refined correction factor for biogenic emissions are generally small or negligible when compared to the method uncertainties. We added a similar comment and the related reference.

3) page 6, line 2: FDMS usually operates at 30 ∘ C. Any comment about possible mass losses (e.g. nitrate?)

We operated the FDMS-TEOM in an air conditioned cabin at a temperature of 21 °C. The FDMS instrumental extension of the TEOM system was developed specifically to account and correct for semi-volatile PM components (see page 5, line 42 – page 6, line 1 of the original MS). Its correct operation was confirmed several times earlier by comparing the filter-based $PM_{2.5}$ mass data (collected outside the BpART platform on Nuclepore filters at ambient air temperature) to the mean obtained from the on-line FDMS-TEOM data set averaged to the corresponding sampling time interval.

4) page 6, line 6 and 9-10. Basing on Drinovec et al. (2015), Atmos. Meas. Tech., 8, 1965–1979 (page 1970) I guess that 16.6 m2/g is the mass extinction (not absorption) coefficient: Aethalometer assumes proportionality between babs and batn through a proportionality factor C - Weingartner et al, (2003), J. Aerosol Sci., 34, 1445–1463. The mass absorption coefficient historically considered by Aethalometer @ 880 nm is 7.77 m2/g (Drinovec et al. (2015)). If a-priori assumption has to be applied to convert PAS measurements into BC concentration maintaining equivalence with what performed in the Aethalometer, then 7.77m2/g is the value to be extrapolated to 1064 nm for application to PAS measurements (please evaluate also the following comment).

We agree that the expression mass absorption coefficient used in the original MS is not completely the correct expression. It represents more a "specific attenuation parameter" or "mass attenuation cross section" (Hansen et al., J. Geophys. Res. 110, D18104, doi:10.1029/2005JD005776. 2005; Drinovec et al., Atmos. Meas. Tech., 8, 1965–1979, 2015) since it is related more to filter artefacts than the differences between the extinction and

absorption coefficients in highly absorbing carbonaceous material. We changed the wording accordingly. The a-priori chosen conversion factor for deriving BC mass concentration from measured optical absorption strongly affects the results. The manufacturer suggested the conversion factor (before a posteriori data correction) of 16.6 $m^2/g$ at a wavelength of 880 nm in earlier models of the AE (such as the AE16, AE22, and AE31). Nevertheless, a conversion factor of 7.77 $m^2/g$ at the same wavelength was proposed for the new AE generation, which adopts automatically the dual spot method (Drinovec et al., Atmos. Meas. Tech., 8, 1965–1979, 2015). We appreciate this comment and modified the data in Tables 1 and 2, in Fig. 1, and related parts of the text at several places in accordance with this.

5) page 6, lines 18-23: the comparisons between optical EBC (from AE or PAS) and thermal EC are strongly affected by the assumption on absorption/extinction coefficients chosen a-priori. As such values are site/season/composition dependent, the parallel availability of optical (batn/babs) and thermal (EC) measurements could be better exploited to provide information on BC mass absorption coefficients in Budapest;

As argued in the answer to comment Aspects requiring attention 4, all the requested information is to be supplied in the MS which is under preparation and to which we refer now in a more direct way in the present paper.

6) page 7, line 37-38: in the cited works, SFUs are reported to have a cut-off diameter of 2 μm. Did the authors change e.g. nuclepore filter pore size or flow-rate to modify the size cut? Different cut-off size could partially justify differences, as well as the use of different instrumentation (i.e. SFU vs. TEOM) or the improvement of combustion technologies in the last 15 years, leading to smaller particles formation.

The Gent-type stacked filter unit (SFU) sampler was indeed designed and realised by one of the co-authors of the present paper (W. Maenhaut) for collecting PM size fractions with nominal aerodynamic diameter (AD) of 10–2 and <2 μm. The cut-point of 2 μm results from drawing air through the top (coarse) Nuclepore filter with 8 μm pore size at a face velocity of 16.6 L/min. The separation between the two size fractions is, however, not very sharp (which is in contrast with the steep impaction collection curve of inertial impactors used for collecting $PM_{2.5}$ or $PM_{10}$). Furthermore, according to P.K. Hopke, the cut-point between the two size fractions in the Gent SFU is rather at 2.2 μm than at 2 μm (P.K. Hopke, personal communication to W. Maenhaut). Thus, the cut-point of 2 μm is rather approximate. In many published papers, in which the Gent SFU was used, including many papers co-authored hy P.K. Hopke and some co-authored by W. Maenhaut (e.g., the paper by Putaud et al. (2010) to which reference is made in our manuscript) the cut-off value between the two size fractions was reported to be 2.5 μm.

In any case, several chemical species have a saddle point in their mass size distribution around the 2–2.5 μm diameter region, so that the difference between their masses in $PM_2$ and $PM_{2.5}$ is virtually negligible. We and the colleagues at the National Air Quality Network in Budapest observed a continuous shift in the mean $PM_{fine}/PM_{coarse}$ mass ratio to larger values in central Budapest during the last 20 years with several collection devices and on-line instruments (e.g. Salma and Maenhaut, Environ. Pollut., 143, 479–488, 2006). This is likely related to some regulatory measures, improvement of combustion technologies and other changes after the economical transition in Hungary, and it does not seem to be influenced by the experimental or sampling method. We amended the text to express this tendency.

7) page 10, lines 9-15. Lignite contribution and its high LVG/MAN does not allow reliabile softwood/hardwood quantification. I suggest to completely remove this paragraph.

The sentences were removed as requested. (We think that this comment concerned page 11, line 9–15 of the original MS.)

8) page 11, line 27: "the fC values of the back filters were individually taken into account for the front filters". Please better explain the procedure for front filter correction.

The amounts of C on the front and back filters were first corrected to the sample preparation yield using the TC data of the OC/EC analysis with the laboratory TOT instrument, and then each amount of C-14 and C-12 on the back filter was subtracted from the corresponding isotope amounts on the front filter in order to obtain the corrected C-14 and C-12 amounts and their fraction. This sampling artefact seems to be particularly important for low-volume samplers. The correction method is now explained more explicitly.

9) page 12, line 5: from Bernardoni et al. (2013) I guess the mentioned value was obtained elsewhere (Bernardoni et al, 2011, Sci Total Environ, 409, 4788–4795)

We adopted one of the EC/OC ratios for wood burning from the cited paper (Bernardoni et al., J. Aerosol Sci., 56, 88–99, 2013, p. 95). Since the value was indeed obtained in an earlier study, we also cite now its original literature source, as requested.

10) page 12, figure 3 and lines 9-14: I don't think the chart correctly represents information flow described in the text. Chart information flow makes argue that factor f2 and f3 are needed to determine EC-BB and OC-BB from TC-C and OC-C. This interpretation would be nonsense, as no proportionality is expected between EC-BB and TC-C, as well as between OC-BB and OC-C (due to at least an independent source - BIO contributing to OC-C). Nevertheless, the text shows that the source apportionment follows a completely different path. If I correctly interpreted the text, it should be evidenced in the chart that input data are obtained from 14C

information (TC-C), and LVG (OC-BB and EC-BB thanks to emission ratios). f2 and f3 are then a by-product (by ratios or subtractions) of these quantities and do not have anything to do with "mass balance equations", opposite to what reported at line 10. Then, I guess from the text that EC-FF is derived as difference between total EC and EC-BB. EC-FF is then exploited to obtain the factor f4. I suggest to completely re-think the chart, including external inputs and their role in the source apportionment scheme. Arrows should start from input data and point to calculated quantities. Furthermore, corresponding text (lines 9-14, page 12) is quite obscure in its current form, and it should be re-written to better evidence the real information flow.

The related part is extensively rewritten, and Fig. 3 is also extended to express the calculation flow more precisely and according to the request of the Referee. See also the answer to Aspects requiring attention 1.

11) page 12, lines 14-17: I suggest to mention also a disadvantage of the proposed model compared to the direct determination of 14C in OC and EC fraction, mainly impacting on EC source apportionment: in the case of the proposed model, EC source apportionment requires assumption on wood burning LVG/EC emission ratio, whereas if 14C is directly measured in the EC fraction, the apportionment is straightforward.

A new paragraph dealing with these issues is now added to the text. We also ask to consider the closing remarks of the present Response.

12) page 13, line 25: "The importance of BB sources, FF combustion and biogenic sources for the PM2.5 mass". Please note that the model does not apportion the total contribution of these sources to PM2.5, but only the contribution of carbonaceous species emitted by these sources (e.g. high emissions of K+ from BB combustion are expected to impact on PM2.5, but they are - of course - completely neglected by this source apportionment approach). Their total relative impact to PM2.5 is for sure higher, and the proportions among them can be different.

We appreciate this comment, and a brief discussion is now added in the text.

13) page 13, line 31: please, add some considerations on estimated uncertainties related to the model.

The discussion is now added.

Page 14, line 16: please add considerations on the possible impact of secondary compounds related to BB emission on OC-BIO estimates

A brief extension on the effects of the secondary compounds related to the BB on the $OC_{BIO}$ is now implemented.

14) page 14, line 19 and lines 26-27: at line 19 the authors find correlations between NOx (assumed as markers for traffic emissions) and EC-FF, whereas no significant correlation is found between NOx and OC-FF. I think this observation supports considerations 2 and 3 at

lines 26, 27. Nevertheless, an important role can be played also by secondary OC-FF, justifying disentanglement between primary species from FF combustion and OC-FF. It should also be noticed that to avoid effects due to different repartition of NO vs. NO2, NOx concentration should be better represented in ppb for the evaluation of more robust correlation coefficients.

A brief comment on the role of secondary $OC_{FF}$ is now added. The concentration of $NO_x$ was measured at some distance from the actual research station, and its 12-h mean concentration was utilised to calculate its correlation coefficients with the carbonaceous chemical species. This fact may play a larger limiting role than the repartitioning between NO and $NO_2$.

**Minor comments**

1) page 3, line 39: what is TEOM size-cut?

The upper size-cut for all aerosol sampling inlets (except for the DMPS which does not have any) was 2.5 μm. It is given on page 4, line 4–6 of the original MS.

2) page 4, line 8-9: don't agree with the sentence. EUSAAR_2 was initially developed for background sites (see end of introduction in Cavalli et al., 2010 Atmos. Chem. Phys., 11, 10193–10203). Its application has effectively been extended to urban sites. Nevertheless, concerns were posed in the literature on its use in urban areas (e.g. Piazzalunga et al., Atmos. Chem. Phys., 11, 10193–10203, 2011)

The sentence was reformulated to include the issues raised by the Referee.

3) page 4, line 11: EUSAAR_2 protocol is longer than 15 minutes. Was it somewhat modified? Please, report, effective protocol

The overall sampling and analytical protocol took 3 hours. We modified the text to express this explicitly.

4) page 4, line 36: why did the authors use different thermal protocols for on-line and off-line instrumentation?

The thermal protocol for the laboratory OC/EC analyser was selected for comparative reasons (to preserve the possibility of the inter-comparison over years) since it has been also employed in our earlier studies in Budapest (page 4, lines 37–39). The related sentence was extended to make this reasoning more evident.

5) page 8, Table 2: I expected mean values for PM10 being the sum of mean values for PM2.5.and PM10-2.5. It is not so. Is it a time-averaging problem? Or can it be due to nonparallel sampling (e.g. missing values in only one of the two samplers were considered in the average?)

The mean value of the on-line $PM_{2.5}$ mass derived for a specific sampling time interval was first added to the corresponding filter-based $PM_{10-2.5}$ mass on a sample-by-sample basis to obtain the $PM_{10}$ mass data set, and then the overall mean value for the latter data was calculated. The mean value of the summed masses (thus the mean of the $PM_{2.5-10} + PM_{2.5}$ data set) doesn't necessarily and exactly equal the sum of the mean mass values (thus the mean of the $PM_{2.5-10}$ data set + the mean of the $PM_{2.5}$ data set). The former data treatment is the correct method considering the dynamic character of atmospheric concentrations, uncertainties of the data, and the propagation law of errors. This was actually utilised in the MS.

6) page 9, line 3: LVG is not a substantial part of OC (as can be seen in figure 1). The observation can indicate an important contribution of LVG source (BB) also to other OC species.

The sentence was reformulated to express the relationship between the LVG and OC more precisely. The argument concerned the temporal relationship between them, and not their amounts.

7) page 9, figure 1: please plot EC and BC (if maintained) on a secondary axis to better allow appreciating modulations

The difference between EC_TOT and BC_PAS became more evident by adopting the new conversion factor for BC (see the answer to Major comment 4).

8) page 10, line 1: no % is needed in OC/EC ratio at the beginning of the line. Furthermore, I suggest to comment this datum which gives indication of strong secondary organic aerosol formation.

The typing mistake was corrected, and the comment was adopted.

9) page 11 line 29: the authors relate the need of sampling artefact correction to the total carbon load on a filter ("1 mg"), and not to the mass/area load. Can they justify this?

The sample preparation for and analytical data from the AMS apply to the total amount of C on the portion of the filter analysed, and not to the surface density. The sentence was extended to express this more precisely.

Finally, we think that the comments of the Referee helped us to improve the MS. We really appreciate this. We would also like to emphasize that the major message of the MS lies in the new pragmatic coupled radiocarbon-LVG apportionment scheme itself, which allows to assess the contribution of the major carbonaceous species with a reasonable uncertainty, and without coupling of thermal or separation methods with an AMS for rather small amounts of samples, which may not be accessible for many research groups.

Imre Salma
for all co-authors

---

## Author Comment (AC2) · 23 Sep 2017

**Response to Referee #2**

The authors thank Referee #2 for his/her detailed and valuable comments to further improve and clarify the MS. We have considered all recommendations, and made the appropriate alterations. Our specific responses to the comments are as follows.

**Comments**

1) Main added uncertainty to the classical method of radiocarbon analysis with the split of EC and OC before the analysis is included in predefined value of ECBB to OCBB ratio. Nevertheless, the authors do not comment uncertainty connected with this value at all. Moreover, although they found that one of the samples was probably contaminated by artificial 14C release, they omitted to discuss a possibility of such influence on the other samples.

A separate discussion part of the uncertainties connected with the apportionment method is now added to the text with a sensitivity calculation. We also like to indicate that the contamination occurred usually (and predominantly) for the back filters, and therefore, it may have an effect (be it limited) on the final uncertainty if it is not properly realised.

2) Finally, although the authors say that their OCBIO is in line with Simpson et al. (2007) the opposite is true due to different definition of fractions. The observed result was 0.17 µ g/m3 of OCBIO in K-puszta for winter (primary biogenic particles), which was roughly 0.2% of TC in PM10 and ca 20% of TC was attributed to non-fossil SOA. Therefore the reviewer suggests to call the last OC fraction same way (non-fossil SOA), also with regard to the text on page 12, line 44.

We decided finally to remove this part of the sentence to avoid discussions on primary and secondary particles, and indicated briefly the alternative attitude in the text.

3) The authors also spend quite a lot of space describing aethalometer and other on- line instrumentation results but they did not use an opportunity to test their method with aethalometer source apportionment method.

The authors regret that they did not indicate to the handling editor and the potential referees at the submission stage that another MS, which deals with and evaluates the data from the on-line measurements obtained by the AE, PAS and DMPS in focus, is under preparation. This other MS will include the evaluation methods referred to by the Referee, thus the so-called Aethalometer model (the wavelength dependence of the optical absorption coefficient) for both the AE (Sandradewi et al., Environ. Sci. Technol., 42, 3316–3323, 2008; Sandradewi et al.,

Atmos. Environ., 42, 101–112, 2008; Favez et al., Atmos. Environ., 43, 3640–3644, 2009; Favez et al., Atmos. Chem. Phys., 10, 5295–5314, 2010) and PAS–DMPS (Ajtai et al., Atmos. Environ., 122, 313–320, 2015) data sets. In addition, a new approach of combining the PAS, DMPS (number size distribution) and AE (soot) data sets into one model is also planned. Comparing the results, advantages and limitations of the filter-based approach (in the present paper) and several optical-related source apportionment methods based on the on-line data sets will be one of the key points of this other MS. We decided to split the whole matter into two parts (with a natural division line between the collection of aerosol samples and on-line measurements) because we consider that their joint presentation 1) could be too lengthy and perhaps even too complex and likely could have a too broad scope, and therefore, 2) it would detract the attention from the new apportionment scheme proposed in the present MS, and 3) our goal is not to prepare a comparative MS on a selection of apportionment methods. We indicated now these additional arguments, motivations and aims very briefly in the text. Nevertheless, it seems plausible and advantageous to describe the experiments of the aerosol sample collection and measurement campaign together in order to have a more comprehensive overview on the aerosol campaign and resulting analytical data sets as a whole, and to assist the future MS to focus the attention specifically on modelling issues and data validation.

4) page 4, lines 33-34: No blank uncertainty is given for mass on Nuclepore and quartz filters

The blank uncertainties are now added to the means. In an earlier study, the role and typical field blank uncertainties for Nuclepore, Teflon and quartz filters in the SFU sampler, which was also utilised in the present work, were determined from a larger number of sample sets, and they were discussed extensively (Salma et al., Atmos. Environ., 38, 27–36, 2004). It was concluded, for instance, that the uncertainty of the Nuclepore samples allows one to determine the PM mass more reliably than is the case with quartz fibre filters, and that the latter substrate is subject to several sampling and handling uncertainties. We refer now to this paper as well.

5) page 6, line 1-2: It seems unusual to combine online PM2.5 mass with offline filters (PM10-2.5) to construct PM10, especially when PM2.5 mass from filters is available even from the same filter pack. Moreover, fine fraction from Ghent SFU is rather PM2 than PM2.5 (Hopke et. al 1997) and therefore coarse fraction will be probably PM10-2 instead of PM10-2.5.

We decide to combine the $PM_{10-2.5}$ mass obtained from the Nuclepore filters with the corresponding mean on-line FDMS-TEOM $PM_{2.5}$ mass because the uncertainty for the $PM_{2.5}$ mass determined from the 12-h exposed fine quartz filters was considerably larger than that for

the mean on-line data. See also the answer to comment 4, and the paper referred there. The Gent-type stacked filter unit (SFU) sampler was indeed designed and realised by one of the co-authors of the present paper (W. Maenhaut) for collecting PM size fractions with nominal aerodynamic diameter (AD) of 10–2 and <2 μm. The cut-point of 2 μm results from drawing air through the top (coarse) Nuclepore filter with 8 μm pore size at a face velocity of 16.6 L/min. The separation between the two size fractions is, however, not very sharp (which is in contrast with the steep impaction collection curve of inertial impactors used for collecting $PM_{2.5}$ or $PM_{10}$). Furthermore, according to P.K. Hopke, the cut-point between the two size fractions in the Gent SFU is rather at 2.2 μm than at 2 μm (P.K. Hopke, personal communication to W. Maenhaut). Thus, the cut-point of 2 μm is rather approximate. In many published papers, in which the Gent SFU was used, including many papers co-authored hy P.K. Hopke and some co-authored by W. Maenhaut (e.g., the paper by Putaud et al. (2010) to which reference is made in our manuscript), the cut-off value between the two size fraction was reported to be 2.5 μm. In any case, several chemical species have a saddle point in their mass size distribution around the 2–2.5 μm diameter region, so that the difference between their masses in $PM_2$ and $PM_{2.5}$ is virtually negligible.

6) page 6, line 7 and elsewhere: The term "data lines" should be probably "data points"

The data files recorded by the on-line instruments usually consist of many rows/lines of data. A row contains usually a date/time stamp in the first columns, and various other measured quantities (e.g. optical absorption coefficients at several wavelength, or mass concentration and reference mass concentration) in the following columns from which the final data of interest are calculated. By writing the expression data rows, we referred to this structure of the collected data file.

7) page 12, Fig 3.: The scheme is quite unclear, needs improvement.

The apportionment scheme is now described more explicitly. We clarify all input data that are necessary for the apportionment and their actual entrance point into the scheme more systematically in both the text and Fig. 3. The arrows in Fig. 3 were originally selected in such way that they indicate the pathways and their major steps in the data treatment flow starting from the measured TC concentration toward the assessed end quantities of the five major carbonaceous species, i.e. $EC_{FF}$ and $OC_{FF}$, $EC_{BB}$ and $OC_{BB}$, and $OC_{BIO}$, which is considered

helpful in overviewing the idea of the proposed new approach for readers who are not very familiar with dedicated with source apportionment.

8) page 14, line 5: The term "tendentious" seems not properly used in the text several times (three times on this page), please check.

The term "tendentious" is removed from the text and it is replaced by other wording.

Imre Salma
for all co-authors